# Middle ratings rise regardless of grammatical construction: Testing syntactic variability in a repeated exposure paradigm

**J. M. M. Brown**[1]*, **Gisbert Fanselow**[1,2], **Rebecca Hall**[3], **Reinhold Kliegl**[1,4]

**1** SFB 1287 Limits of Variability in Language, University of Potsdam, Potsdam, Germany, **2** Department of Linguistics, University of Potsdam, Potsdam, Germany, **3** Department of Psychology, University of Potsdam, Potsdam, Germany, **4** Department of Sport Sciences, University of Potsdam, Potsdam, Germany

* jmmbrown@cantab.net

**Data Availability Statement:** All data and scripts are available from the Open Science framework database: https://osf.io/ge2db/.

## Abstract

People perceive sentences more favourably after hearing or reading them many times. A prominent approach in linguistic theory argues that these types of exposure effects (satiation effects) show direct evidence of a generative approach to linguistic knowledge: only some sentences improve under repeated exposure, and which sentences do improve can be predicted by a model of linguistic competence that yields natural syntactic classes. However, replications of the original findings have been inconsistent, and it remains unclear whether satiation effects can be reliably induced in an experimental setting at all. Here we report four findings regarding satiation effects in *wh*-questions across German and English. First, the effects pertain to zone of well-formedness rather than syntactic class: all intermediate ratings, including calibrated fillers, increase at the beginning of the experimental session regardless of syntactic construction. Second, though there is satiation, ratings asymptote below maximum acceptability. Third, these effects are consistent across judgments of superiority effects in English and German. Fourth, *wh*-questions appear to show similar profiles in English and German, despite these languages being traditionally considered to differ strongly in whether they show effects on movement: violations of the superiority condition can be modulated to a similar degree in both languages by manipulating subject-object initiality and animacy congruency of the *wh*-phrase. We improve on classic satiation methods by distinguishing between two crucial tests, namely whether exposure selectively targets certain grammatical constructions or whether there is a general repeated exposure effect. We conclude that exposure effects can be reliably induced in rating experiments but exposure does not appear to selectively target certain grammatical constructions. Instead, they appear to be a phenomenon of intermediate gradient judgments.

## Introduction

### Changing judgments

Speaker judgments about the well-formedness of sentences form an important source of evidence for evaluating linguistic theories. For more than two decades, linguists have increasingly

**Funding:** Funded by the Deutsche Forschungsgemeinschaft (DFG, German Research Foundation) – Project ID 317633480 – SFB 1287, Project C01. Principal Investigators: GF and RK. URL of the funder is: https://www.dfg.de/ The funders had no role in study design, data collection and analysis, decision to publish, or preparation of the manuscript.

**Competing interests:** The authors have declared that no competing interests exist.

used experiments to collect acceptability judgments in a systematic way (e.g., Schütze [1], Cowart [2]). Measuring acceptability in a systematic way has demonstrated that judgments of certain constructions change when this construction is repeatedly presented to the same participant. Such changes are documented as early as Nagata [3] and Carroll [4]. The first major treatment in linguistics is Snyder [5], where he argues that constructions that improve under repeated exposure form a natural syntactic class (these constructions "satiate" as he calls it, attributing the introduction of the term into syntax to Karin Stromswold). In other words, independently motivated linguistic and psycholinguistic properties determine whether a construction can or cannot satiate. Therefore, the availability of satiation for a given construction could be taken as an indication that this construction has a particular linguistic or psycholinguistic property. Which property makes sentences prone to satiation remains unclear. Explanations in the literature range from structural properties within generative syntax (Snyder [5]: 579 classical "subjacency" effects in the sense of Chomsky [6]) to sentence processing difficulty (as in, e.g., Chaves & Dery [7, 8]) to issues of comprehension fluency (e.g., Zervakis & Mazuka [9]).

The outcome of the three experiments reported here suggests a different perspective (first taken in a different form by Nagata [3]): satiation does not discriminate between construction types on the basis of some abstract property. Rather, satiation indiscriminately affects all types of sentences within a certain zone of judgments. "Zone of judgments" refers to an interval in the judgment space between "fully acceptable" and "fully unacceptable" such that satiation affects all structures rated within the interval, and none of the structures rated outside the interval. It is plausible to relate this zone to the phenomenon of "intermediate" acceptability, but it remains difficult to anchor this zone with precise acceptability values on, for example, an n-point scale, because specific grades of acceptability are co-determined by various orthogonal design factors in the experiments (such as the nature and quality of the fillers and the target-filler ratio or the range of the n-point scale). Nevertheless, for this study we consider the "middle" zone as the space between the upper and lower 25% of the judgment space (i.e., between 2.75 and 5.25 on a 7-point scale). We note again though that our empirical claim does not so much focus on specific values, but rather on the pervasive nature of satiation within such a domain. Therefore 2.75 and 5.25 are not absolute upper and lower thresholds but rather serve the purpose of comparing satiation effects (or the lack thereof) across target and filler conditions, and across experiments. More likely than not, the upper and lower end of the zone will themselves be of a gradient nature.

We draw our conclusions with reference to ratings of a set of standardised filler groups based on those developed by Featherston and colleagues (Featherston [10], Gebrich et al. [11]) for the purpose of calibrating syntactic judgments: each group comprises a set of constructions that are diverse syntactically but share a specific value in the judgment space. All filler groups turn out to satiate when their judgment value falls between the highest and the lowest conditions of the core experiment affected by syntactic satiation in our experiments.

In the remainder of the introduction, we review some of the literature to highlight the diversity of approaches, both in terms of syntactic manipulations and experimental designs. This review yields a somewhat incoherent and sometimes contradictory profile of results in this field of research. Against this background, we provide the reasons motivating our use of superiority violations and their examination in German and English and the role of fillers in this context.

The literature on syntactic satiation effects is already quite large, and has been summarised and discussed in detail in Snyder [12]. Experiments have been quite diverse methodologically, and differ in how they identify syntactic satiation effects. In line with this diversity, the results are quite variable–despite the fact that much of the literature focuses on the same set of

"classical" constructions first investigated for syntactic satiation by Snyder ([5]: 576). These constructions are illustrated in (1) in which a gap created by moving some phrase is marked with (t) for 'trace'. In (1a), the argument *who* moves from within a want-for clause to the front of the sentence. In (1b), *who* is extracted from within a clause introduced by *whether* (*wh*-island). In (1c), who is moved to the front of the sentence across *that (that-trace-effect)*. In (1d), *what* is extracted from within a complex subject (extraction from a subject island). In (1e), *who* is extracted from within a complex noun phrase, violating the Complex Noun Phrase Constraint. In (1f), *who* is extracted from within an adjunct (movement out of an adjunct island), and in (1g), *how many* is extracted from within the noun phrase *how many books* without pied-piping the rest of its constituent (violation of the Left Branch Condition).

1. Classic satiation constructions

   a. *Want-for*: *Who does John want for Mary to meet (t)?

   b. *Whether-island*: *Who does John wonder whether Mary likes (t)?

   c. *That-trace*: *Who does Mary think that (t) likes John?

   d. *Subject island*: *What does John know that a bottle of (t) fell on the floor?

   e. *Complex NP island*: *Who does Mary believe the claim that John likes (t)?

   f. *Adjunct island*: *Who did John talk with Mary after seeing (t)?

   g. *Left branch*: *How many did John buy (t) books?

   [Snyder [5]: (2a), 576]

Relatively few studies investigated other constructions of English (e.g., Sprouse [13], Goodall [14], Zervakis & Mazuka [9]). In addition, there is some limited amount of work on different languages. Goodall [14] reports satiation effects for some constructions (e.g., questions without subject verb inversion) in Spanish but no such effects for, for example, Spanish counterparts of (1d), (1e) and (1f). In Danish, satiation effects are reported for sentences with long movement, and no satiation for sentences with short movement or no movement at all or sentences with doubly-filled specifiers (Christensen, Kizach & Nyvad [15]).

Table 1 summarises the outcome of a sample of satiation studies with (1) or a subset thereof as a point of reference.

The main observation emerging from Table 1 is how mixed previous results have been. The only consistent results are the negative ones for left branch condition violations (1g). Some studies find satiation effects for certain of the constructions but others do not. Probably, this variability is at least partially due to differences in method.

When we focus on the idea that syntactic satiation is related to a "zone of acceptability", we again observe mixed results in the literature. Christensen, Kizach, and Nyvad [15] found a positive correlation of order of presentation and judgments in two experiments focusing on Danish *wh*-movement that are compatible with our hypothesis. In their Experiment 1, all constructions of intermediate acceptability (2.43–3.66 on a 5-point scale) showed the satiation effect but none of the sentences with more extreme values. This is in agreement with our 25% criterion ranging from 2.25 to 3.75 on a 5-point scale. Experiment 2 sheds a slightly different light on the hypothesis. It was partially identical to Experiment 1, but (among other changes) all structures (e.g., short movement) with an acceptability value above the saturation range of Experiment 1 were removed. Again, a continuous segment of the rating scale (3.86–4.36) underwent satiation, but the high upper boundary (4.36 on a 5-point scale) does not so much reflect satiation for items with near-perfect acceptability as the absence of more well-formed

**Table 1. Summary of satiation effects by previous study and construction.**

| | Want-for | *Whether*-island | *That*-trace | Subject island | Complex NP island | Adjunct island | Left branch condition |
|---|---|---|---|---|---|---|---|
| Braze (2002) [16] | | **yes** | | | | no | |
| Chaves and Dery (2014) [7]: exp. 1 | | | | **yes** | | | |
| Chaves and Dery (2014) [7]: exp. 2 | | | | **yes** | | | |
| Chaves and Dery (2019) [8]: exp. 1 | | | | **yes** (subject gap) | | | |
| | | | | no (object gap) | | | |
| Chaves and Dery (2019) [8]: exp. 3 | | | | **yes** (parasitic gap) | | | |
| | | | | **yes** (object gap) | | | |
| | | | | no (non-parasitic gap) | | | |
| Crawford (2011) [17] | | **yes** | | no | | no | |
| Francom (2009) [18]: exp. 1 | **yes** | **yes** | no | **yes** | no | no | no |
| Francom (2009) [18]: exp. 2 | | | no | **yes** | no | no | no |
| Goodall (2011) [14] | | | no | **no** | **yes** | **no** | no |
| Hiramatsu (2000) [19] | **yes** | **yes** | **yes** | **yes** | no | no | no |
| Snyder (2000) [5] | no | **yes** | no | no | **yes** | no | no |
| Sprouse (2007a) [15] | | no | | no | no | no | |
| Sprouse (2007b) [16]: exp. 1 | no | no | no | no | no | no | no |
| Sprouse (2007b) [16]: exp. 2 | no | no | no | no | no | no | no |
| Sprouse (2007b) [16]: exp. 3 | no | no | no | no | no | no | no |
| Sprouse (2009) [20]: exp. 1–5 | | no | | no | no | no | |
| Sprouse (2009) [20]: exp. 6 | | no | | no | no | no | no |
| Sprouse (2009) [20]: exp. 7 | | no | | | no | no | |

conditions in the experimental material (compared to Experiment 1, the rating of long movement had gone up by 0.7 points). Thus, the comparison of the two experiments illustrates the difficulty, already mentioned above, of defining the zone of "intermediate acceptability" purely numerically in terms of values on an n-point scale. Moreover, the experiments had their empirical focus on contrasts in the length of movement, so it is not obvious that their interpretation can be generalised beyond this domain.

Zervakis and Mazuka [9] tested a more varied set of constructions for satiation effects with a between-subject design. The experimental group rated 5 blocks of 100 experimental sentences on a 7-point scale. The blocks occurred in different orders, such that each of the blocks was rated as the last one by the same number of participants. The control group began by rating 400 control sentences and then judged the acceptability of one of the five experimental blocks. A comparison of the experimental with the control group revealed a significant increase in acceptability for all constructions rated higher than 2.67 by the control group up to a garden path construction rated at 4.56. However, a condition called "grammatical binding" falling well into this interval (3.45) failed to satiate, and the simple filler sentences also went up in their ratings although their initial value was at 6.03 (Zervakis & Mazuka [9]: 513).

There are also proposals (Snyder [5], Goodall [14]) explicitly arguing against the idea that satiability is a by-product of intermediate acceptability. Snyder [5] identified *whether*-island violations (1b) and Complex Noun Phrase Constraint violations (1e) as satiating constructions, but not, for instance, *that*-trace violations (1c), by comparing the number of participants giving more "yes" responses to such constructions in the second half of a categorial acceptability task than in the first half, to those showing the opposite dynamics of judgment. In a second experiment, different participants were asked to rate the same material on a 5-point scale. Two constructions did not satiate (*that*-trace (1c) and subject island violations (1d)) and these

violations were rated in between two conditions that did satiate (*whether*-island violations (1b) and Complex Noun Phrase Constraint (1e) violations).

Snyder's basic finding was replicated in experiment 1 of Francom [18], where *whether*-island, Complex Noun Phrase Constraint (1e), *that*-trace (1c) and subject island (1d) violations turned out to have basically the same level of acceptability (between 30 and 40% positive replies in a categorial judgment task, Francom [18]: 35), while only two of them satiated (*whether*- and subject islands violations). However, when applying a more balanced design, combined with an increase in the number of items per condition, Francom [18] not only observed changes in the acceptability rankings of the constructions (with the ratings of Complex Noun Phrase Constraint violations falling below those of *that*-trace and subject island violations in comparison to what experiment 1 had revealed), but also found satiation effects restricted to subject island violations (Francom [18]:56–58).

Goodall [14] provides another example of the sensitivity of acceptability ratings to local context. For English, he reports a considerable increase in the acceptability of Complex Noun Phrase Constraint violations. However, despite comparable acceptability judgments in the first block, this increase was not seen for *that*-trace violations in the final two of five experimental blocks. Clearly this result is not in agreement with the hypothesis that satiation is a function of the degree of acceptability. In Francom ([18], Experiment 2) violations of the Complex Noun Phrase Constraint did not satiate but subject island violations did. The variability between experiments calls for an investigation of satiation with a more powerful design.

The variability of methods extends also to the intensity of the repeated exposure of the participants to some construction. Zervakis and Mazuka [9] used a block design with a large number of items in a between-subject design; they found quite pervasive satiation effects. We hypothesise that satiation effects are of moderate size and arise only after a considerable amount of exposure.

In summary, the review of a select set of studies above highlights the need for employing a standardised experimental paradigm with the ideal of an equal number of items for all levels of gradient acceptability in the material and a maximum of counterbalancing their order of presentation across the experiment. The experimental design must also aim for large statistical power with a large number of subjects and items, within-subject/within-item experimental manipulations, and a sufficient number of blocks of trials. Such a design is expected to yield evidence on the shape of the satiation function: does it arise quickly and asymptote at or considerably below close-to-perfect acceptability?

## A syntactic case study: Superiority effects

The final topic of the introduction is on the languages and the constructions used in our experiments. As in Goodall [14], we decided to look for satiation effects for a comparable or even identical set of constructions in two languages, rather than only one. If it is the intermediate nature of the judgment rather than specifics of grammatical properties that are responsible for satiation effects, the same or a similar satiation functions should emerge for acceptability ratings in both languages. Conversely, we expect qualitative differences if grammatical properties matter, as demonstrated by Goodall's discussion of why Complex Noun Phrase Constraint violations satiate in English but not in Spanish due to their grammatical differences, in spite of comparable initial acceptability.

There is a tension between the goals of using close-to-identical sentence material in the experiments (changes ideally being restricted to the use of different lexicalisations) and the goal of reducing the impact of construction type by making level of acceptability the foremost criterion for deciding on the material. Our experiments aimed for a compromise on these

incompatible demands by basing the main experiment of targets on constructional similarity and by also including a systematic set of fillers representing six levels of acceptability in both language by different constructions.

English and German are an ideal pair of languages for measuring change in superiority judgments. While they are closely related, and share many constructions in terms of surface appearance, the grammatical analysis of these constructions for the target sentences can be quite different as shown, for example, by Haider [21] and many others. This applies in particular to the grammar of multiple questions, illustrated in (2).

2. a. John knows who saw what.
   b. *John knows what who saw (t).

Multiple questions are well-formed in English when the order of the *wh*-phrases corresponds to their normal linearisation in declarative sentences, as in (2a), in which the *wh*-subject precedes the *wh*-object. The placement of a *wh*-object in front of a *wh*-subject usually leads to a remarkable drop in acceptability (2b), called "superiority effect", as first observed by Kuno and Robinson [22]. While there is some disagreement in the literature, the standard hypothesis in generative syntax is that the unacceptability of (2b) reflects the operation of some *grammatical* principle. which we will call "superiority condition" in this paper without committing ourselves to any of the proposals (see, e.g., Häussler et al. [23], and Fanselow, [24]) for a discussion.

The situation is different in German. Like in English, the acceptability of object initial multiple questions was found to be reduced as compared to their subject-initial counterpart in a number of experimental studies, yet syntactic reasons summarised in Haider [25] have led to a widespread conviction that whatever kind of superiority effect one may observe in German, it does not reflect the operation of a syntactic superiority condition but is due to a "conspiracy" of a number of factors, some of which pertain to the realm of language processing (see Häussler et al. [23], for experiments meant to support this perspective). Thus, superiority effects of German and English may arise from different sources. Furthermore, many experiments have revealed the gradient/intermediate nature of judgments related to the superiority effect (see, again, Häussler et al. [23], for an overview). Multiple questions are also the only domain for which there is a sufficient number of experimental studies comparing English and German to base the present study on (Featherston [10], Häussler et al. [23]). Finally, the presence of satiation effects for multiple questions has been hypothesised in the previous literature (Hofmeister et al. [26]), but systematic studies concerning satiation in multiple questions have not been undertaken so far.

We have good reasons for the choice of the superiority effect, but there are also reasons for deciding against using the "classical" construction set of Snyder [5]. First, there is no counterpart of the *want-for* construction of English in German. Second, German shows regional variation with respect to the status of moving elements out of a complement clause (see, e.g., Fanselow & Weskott [27]). Long movement in the experimental items may thus incur an unwanted impact of sociolinguistic evaluation of the appropriateness of using regional variety examples on judgments of "formal" well-formedness that one would not see in the English counterpart. Rather than trying various options for fixing this problem, we decided to take recourse to short inner-clausal movement and work on acceptability differences for which we are not aware of any dialectal or sociolinguistic variability.

For the second dimension of our experiment, namely using fillers representing different levels of (gradient) acceptability, we used constructions identified using a norming study in Featherston [10], and Gebrich et al. [11]. Based on a set of carefully implemented experiments, they identified five sets of structurally diverse sentences which occupy constant relative points

in the judgment space and can thus be used as "calibrators" anchoring these points in the judgment space. Examples for some of the intermediate English levels are illustrated below, showing that there is no (obvious) property they have in common but the level of acceptability.

Level C:

Hannah hates but Linda loves eating popcorn in the cinema.

Most people like very much a cup of tea in the morning.

The striker must have fouled deliberately the goalkeeper.

Level D:

Who did she whisper that had unfairly condemned the prisoner?

The old fisherman took her pipe out of mouth and began story.

Which professor did you claim that the student really admires her?

Level E:

Historians wondering what cause is disappear civilisation.

Old man he work garden grow many flower and vegetable.

Student must read much book for they become clever.

There is a necessary between-language variability for the fillers because the German material is not based on structural similarity but on level of acceptability. Thus, German level C contains (among other material) unusual binding constellations. The acceptability of the example for level D is problematic because the sentence does not respect a leftward placement rule for unstressed pronouns. Two examples are given below. Each level contains three constructions as for English.

Level C:

Ich habe dem Kunden sich selbst im Spiegel gezeigt.

I have the client himself.REFL himself in.the mirror shown

*I showed the client himself in the mirror.*

Level D:

Der Komponist hat dem neuen italienischen Tenor es zugemutet.

the composer has the new Italian tenor it expected.of

*The composer expected it of the new Italian tenor.*

Between-language differences in acceptability ratings for the different levels of filler sentences may pose problems of interpretation for results, but if fillers show a similar behaviour in the two languages, this is most likely due to their position in the judgment space.

## Experiment 1: Rating of German sentences

Experiment 1 tests the effect of repeated exposure in German indirect multiple *wh*-questions in subordinate clauses. Targets crossed whether they were subject-initial or object-initial (the latter case tests for possible superiority effects) and whether there was an animacy congruency between subject and object *wh*-words, yielding a 2 x 2 design. Thus, each target sentence was available in four conditions as shown in (3).

3. German target conditions

a. Condition 1: subject-initial; matching animacy (*wer-wen*); well-formed in German
   Keinesfalls wusste die Haushälterin genau, **wer** bei der Gartenfeier **wen**
   certainly.not knew the housekeeper exactly who by the garden.party who
   ständig angesehen hat.
   continuously looked.at had
   *The housekeeper certainly did not know exactly **who** had kept on looking at **who(m)** at the garden party.*

b. Condition 2: subject-initial; mismatching animacy (*wer-was*); well-formed in German
Keinesfalls wusste die Haushälterin genau, **wer** bei der Gartenfeier **was**
certainly.not knew the housekeeper exactly who by the garden.party what
ständig angesehen hat.
continuously looked.at had
*The housekeeper certainly did not know exactly **who** had kept on looking at **what** at the garden party.*

c. Condition 3: object initial; matching animacy (*wen-wer*); not fully well-formed in German
Keinesfalls wusste die Haushälterin genau, **wen** bei der Gartenfeier **wer**
certainly.not knew the housekeeper exactly who by the garden.party who
ständig angesehen hat.
continuously looked.at had
*The housekeeper certainly did not know exactly **who** had kept on looking at **who(m)** the garden party.*

d. Condition 4: object initial; mismatching animacy (*was-wer*); well-formed in German
Keinesfalls wusste die Haushälterin genau, **was** bei der Gartenfeier **wer**
certainly.not knew the housekeeper exactly what by the garden.party who
ständig angesehen hat.
continuously looked.at had
*The housekeeper certainly did not know exactly **who** had kept on looking at **what** at the garden party.*

The first factor manipulates the superiority/crossing movement variable: conditions 1 and 2 contain two *wh*-phrases, and the subject precedes the object. These conditions therefore illustrate instances where a superiority effect cannot arise, and act as controls against which to compare potential superiority effects in the object-initial targets. Conditions 3 and 4 contain the crucial manipulation: In these conditions, the object *wh*-phrase is placed in front of the subject *wh*-phrase, meaning that a superiority effect could arise.

The second factor manipulates an extra-grammatical feature, namely the animacy of the *wh*-phrases. Thus, subject and object match in animacy in two sentence types, namely (3a) and (3c), while the other two conditions, namely (3b) and (3d) have subjects and objects that do not match in animacy. We start here from the assumption that differences in subject and object animacy increase the well-formedness of crossing movement, see Fanselow et al. [28] and Häussler et al. [23].

As described above, ratings of target sentences are to be analysed not only with respect to the 2 x 2 experimental manipulations (and how they change across the blocks of the experiment), but also in the context of a calibrated set of six types of filler sentences [3]. The first five gradations of fillers varied from (A) "completely well-formed" to (E) "almost not well-formed". We included also a new sixth level (F) "uninterpretable and unacceptable" as a clearly ungrammatical level. Examples are provided under Material below.

Finally, with respect to methodology, we go beyond past designs in this area with a within-subject/within-item counterbalancing scheme built around six blocks of 72 items and a multiple of 24 subjects. As far as we could determine, past research employed only the usual counterbalancing measures for experimental conditions in the theoretical focus. Thus, applied to our design, the four instances of each target sentences are presented equally often in the experiment but such that every subject rates only one instance of each target sentence and rates the same number of items in each of the four conditions. However, in a satiation experiment this

is not enough. As items vary in acceptability we must also ensure that they appear equally often in each of the six blocks of the experiment while respecting the usual constraints. In other words, counterbalancing is extended to a 2 x 2 x 6 scheme. The same also holds for the six types of filler sentences. As type of filler varies within subject, but between items, each filler is presented equally often in each block and rated once by each subject; they also rate the six types of fillers equally often in each block. As another innovative design feature we imposed the constraint that first-order transitions between the four target conditions and the six filler levels occurred equally often (i.e., we used a Williams design [29]). This counterbalancing optimally controls for item differences in acceptability and increases the signal-to-noise ratio for the detection of satiation effects.

## Method

**Subjects.** A total of 55 students and employees of the University of Potsdam participated in the study. Only the results of German native speakers were included in the evaluation; two participants were excluded due to technical failures. Of the remaining 48 participants, 4 were male and 44 female and ranged between 19 and 40 years of age, with an average age of 24 years. Recruitment was carried out via flyers distributed on campus, internet advertisements on various platforms, and an e-mail distribution list from a university experimental laboratory. We did not seek ethics approval because rating questionnaires do not require a full ethics application in Germany. Participation was voluntary and was remunerated with study credit or eight euros. All participants gave written consent and were naive with regard to the questions and objectives of the study.

**Apparatus.** The experiment was conducted in the experimental psychology lab at the University of Potsdam. Three soundproof computer booths were used. The experiment was implemented using the Python software PsychoPy version 1.84.2 [30]. The stimulus material was presented on a computer screen along with a scale labelled from 1 to 7. Underneath each of the numbers was a short description of the degree of well-formedness: 1 "überhaupt nicht wohlgeformt" (not at all well-formed), 2 "fast nicht wohlgeformt" (almost not well-formed), 3 "eher nicht wohlgeformt, als wohlgeformt" (more not well-formed than well-formed), 4 "kann man nicht zuordnen" (cannot be classed as well-formed or ill-formed), 5 "eher wohlgeformt, als nicht wohlgeformt" (more well-formed than not well-formed), 6 "nahezu wohlgeformt" (mostly well-formed), 7 "völlig wohlgeformt" (completely well-formed). Sentence ratings from "1" to "7" were entered on a standard computer keyboard with a German keyboard layout.

**Material.** The material consisted of 120 target quadruples (corresponding to (3) above) and 252 fillers. That is, the ratio of targets to fillers was roughly 1:2. Subjects rated targets in one of four versions (i.e., a total of 480 different sentences were constructed from the 120 targets). A complete list of targets and fillers is provided in the OSF Repository (https://osf.io/ge2db/).

The targets represent indirect multiple *wh*-questions in subordinate clauses. The construction principle yielding the four instances for each of the 120 targets, that is subject/object initial sentence (2) x animacy match/mismatch of subject and object *wh*-words (2), is illustrated in (3) above. In addition, as also shown in the example in (3), half of the target sentences started and the other half ended with an adverb in the main clause; in the subordinate clause there was an adverb after the second *wh*-word. The purpose of adverbs was to make the sentences sound as natural as possible.

The fillers involved six grammatical levels with 42 items each. For the first five gradations from (A) "interpretable and highly acceptable " to (E) "interpretable but less acceptable than (D)", we used calibration sets from [3] to create the fillers. That is, we started from single

examples of each construction at each acceptability level (3 constructions x 5 levels = 15 items) [3], and created 195 more items on the same templates. However, we removed the multiple question construction from level D as they were present in the target material. We added a sixth level (F) "uninterpretable and unacceptable" with 42 items to provide a clearly ungrammatical level.

4. German filler examples (based on the Featherston fillers)

 1. Level A: Interpretable and highly acceptable
In der Mensa essen viele Studenten zu Mittag.
In the canteen eat.3PL many students to lunchtime
*Many students have lunch in the canteen.*

 2. Level B: Interpretable but less acceptable than (A)
Der Kaiser hat dem Fürsten den Maler empfohlen.
The emperor has the prince the artist recommended
*The emperor recommended the artist to the prince.*

 3. Level C: Interpretable but less acceptable than (B)
Ich habe dem Kunden sich selbst im Spiegel gezeigt.
I have the client himself.REFL himself in.the mirror shown
*I showed the client himself in the mirror.*

 4. Level D: Interpretable but less acceptable than (C)
Der Komponist hat dem neuen italienischen Tenor es zugemutet.
the composer has the new Italian tenor it expected.of
*The composer expected it of the new Italian tenor.*

 5. Level E: Interpretable but less acceptable than (D)
Der Waffenhändler glaubt er, dass den Politiker bestochen hat.
The arms.dealer believes he that the politician bribed has
*It is the arms dealer that he believes bribed the politician*

 6. Level F: Uninterpretable and unacceptable
Die Tinte wurde für vergossen.
*the ink was for spilled*

Note that in (4), we follow the convention of providing literal translations for individual items in glosses, followed by the closest meaningful translation in English on a separate line (here in italics). Therefore, although the translations for filler levels (A) to (E) are all fully acceptable and well-formed in English, the German examples themselves from (B) through (E) are not fully acceptable: they decrease in well-formedness between the highest level (A) and the lowest level (F). The lowest filler (F) is intended to be uninterpretable, meaning that there is no meaningful way for all the words to be integrated into the interpretation of the whole sentence, and therefore no full translation of the whole sentence to English. For filler (F), we therefore provide only literal translations for individual items in the gloss, and leave the translation blank.

**Design and counterbalancing.** The counterbalancing scheme used for the experiment differs from past research and was described at the end of the Introduction. Each subject rated 372 sentences (120 targets + 252 fillers) that were distributed across six blocks (i.e., 62 sentences per block; 2 x 2 x 5 = 20 targets and 6 x 7 = 42 fillers). As already described above, each subject rated the 120 targets in only one of the four conditions, and rated five targets in each of the 2 x 2 conditions in each block. Similarly, subjects rated the 252 fillers once and seven fillers

of each of the six types in each block. The counterbalancing scheme also ensured that all targets were rated equally often in their four conditions in each block and that all fillers and filler levels occurred equally often in each block.

Presentation order of targets and fillers also adhered to a Williams design [29]. A Williams design is a type of Latin square design that controls for first-order carry-over effects, ensuring that transitions between the four experimental conditions and the six types of fillers occur equally often. Finally, the item sequence was subject to the following constraints: (a) no immediate repetition of target sentences (i.e., target sentences were bracketed by at least one filler sentence), (b) no immediate repetition of the same experimental condition, and (c) no immediate repetition of the same type of filler sentence.

This counterbalancing scheme requires a multiple of 24 subjects. Twenty-four is a typical sample size for psycholinguistic research, but to increase statistical power we recruited twice the minimal number, that is a total of 48 subjects. Statistical power is also high because all design factors (subject/object-initiality of *wh*-words, animacy-congruency of *wh*-words, block, level of filler, target vs. filler) vary within subject; and only two of them, target vs. filler and level of filler, are between-item factors.

**Procedure.** After providing informed consent and collection of demographic information, subjects were instructed in the well-formedness rating procedure. Specifically, they were asked to judge the well-formedness of the sentences according to spoken—not written—language and to use the full spectrum of the seven-point scale. They practiced the procedure with nine examples spanning the scale of well-formedness, including two of the uninterpretable fillers (filler level F). Then, each subject worked through the six blocks of sentences. At the end of each block, the word *pause* appeared on the screen with instructions to press a button when the participant was ready to continue. Participants could pause in front of the computer at this point. Durations of breaks between blocks was under the subjects' control. Most participants took around an hour to complete the experiment.

**Statistical analysis.** We used the open source software R [31], especially packages *lme4* [32], *tidyverse* [33], *cowplot* [34], *sjPlot* [35], and *broom.mixed* [36] for statistical analyses. Inferential statistics for the analyses of ratings are based on two linear mixed model (LMMs), one only for ratings of targets and one for a joint analysis of targets and fillers.

*LMM for rating of targets*. The LMM for targets included subject and target as crossed random factors and subject- vs. object-initial word order (2), animacy congruency of wh-words (2), and block (6) as fixed factors. The three factors were varied within-subject and within-items. Across blocks we expected satiation of well-formedness ratings, meaning that ratings would increase and eventually reach an asymptote. Therefore, a Helmert contrast [37] was specified for levels of block to capture the point at which the rating no longer changed significantly. To this end the first contrast tested block 1 against the average of blocks 2 to 6, the second contrast the second block against the average of blocks 3 to 6, and so on.

Random effects associated with within-subject or within-item factors potentially give rise to variance components (VCs) and correlation parameters (CPs) of the mean rating and of experimental effects in the random-effect structure of the LMM. They need to be included to guard against false positives, but many VCs and CPs are not supported by the data and, if included, reduce statistical power. We selected an LMM following the strategy outlined in [38]; see also [39]). The significance of fixed effects which are the focus here did not depend on the specifics of the random-effect structure. Details of model selection are documented in the analysis scripts in the OSF repository.

*LMM for targets in the context of fillers*. The second LMM included ratings of both target and filler sentences. We specified a block (first block vs. average of blocks 2 to 6) x type of sentence (10) design; type of sentence comprised four types of target and six types of filler

sentences. The goal of this analysis was to determine for which of the ten sentence types acceptability increased significantly from block 1 to the average of block 2. Therefore, we specified the effect of block as nested within each of the ten levels of sentence type. Model selection, that is determination and inclusion of VCs and CPs followed the strategy outlined in [38]. Due to the complexity of this LMM, we used the JuliaStats/MixedModels.jl package [40] for model fitting and model selection. Details of model selection are documented in the analysis scripts in the OSF repository. Estimates of model parameters and comparative goodness of fit statistics are reported in the supplement. Given the large number of subjects, items, and observations, the usual t-distribution approximates the normal distribution. Therefore, we report test-statistics (estimate / standard error) as z-values and interpret absolute values larger than 2 as significant.

## Results and discussion

**Targets.** The first analysis focuses on the targets involving multiple questions. The four conditions of targets using constructions of (a) "wer-wen", (b) "wer-was", (c) "wen-wer", and (d) "was-wer" for the same sentence frames map onto a 2 x 2 design with the main effects subject-vs-object initial targets (so; a+b-c-d) and animacy congruency of *wh*-words (an; a+c-b-d) as well as the interaction between these two effects (a-b-c+d). In addition, this 2 x 2 design was repeated across six blocks of trials. Fig 1A displays the change in ratings of well-formedness for the four types of targets shown in red; Fig 1A also shows performance for the levels of fillers, but we initially focus only on the targets. The LMM fixed-effect estimates and statistics are provided in the supplement.

As expected, the subject-initial targets were rated as more acceptable than object-initial targets, yielding a significant main effect of order (b = 0.52, z = 8.97). Overall, there was no significant effect of animacy congruency (b = -0.05, z = -1.62), but there was a significant interaction with order (b = 0.13, z = 5.00): there was a very clear preference for the inanimate "was-wer" construction over the animate "wen-wer" construction for object-initial targets, whereas–at least in the first block–the reverse preference held for subject-initial targets.

Indeed, the pattern of change across blocks followed a very simple structure: overall, there was a significant increase in ratings from the first to the average of blocks 2 to 6 (b = 0.31, z = 3.24), and this increase was different for the interaction of order and animacy (b = -0.15, z = -2.60): As is clearly visible in Fig 1, ratings increased less strongly for "wer-wen" compared to the other three constructions (i.e., they did not increase at all), and ratings increased more strongly for "wer-was" constructions compared to the other three conditions. None of the other contrasts defined for the change in ratings across blocks, nor–with one exception–none of the other interaction terms involving these contrasts were significant (all z-values < 2.00). The exception is a marked violation of parallelism between block 4 and 5: only the "was-wer" construction exhibited in increase in rating (b = 0.10, z = 2.34). We consider this interaction as spurious.

There are three main results. First, there is clear evidence for large differences in the judgement of well-formedness of multiple questions with a clear preference for subject-initial than object-initial targets, in line with previous experiments [23, 28, 41] and at the same time there is no reliable evidence that these preferences changed much after one block of trials.

Second, for object-initial targets, mismatched animacy is preferred to matched animacy. The condition with mismatched animacy is almost on par with subject-initial conditions, in line with the findings in Häussler et al. [23] and Fanselow et al. [28]. It is only in the condition with matched animacy that a superiority effect can be seen, and even here the condition is more acceptable than three of the filler levels (including filler levels D and E that are

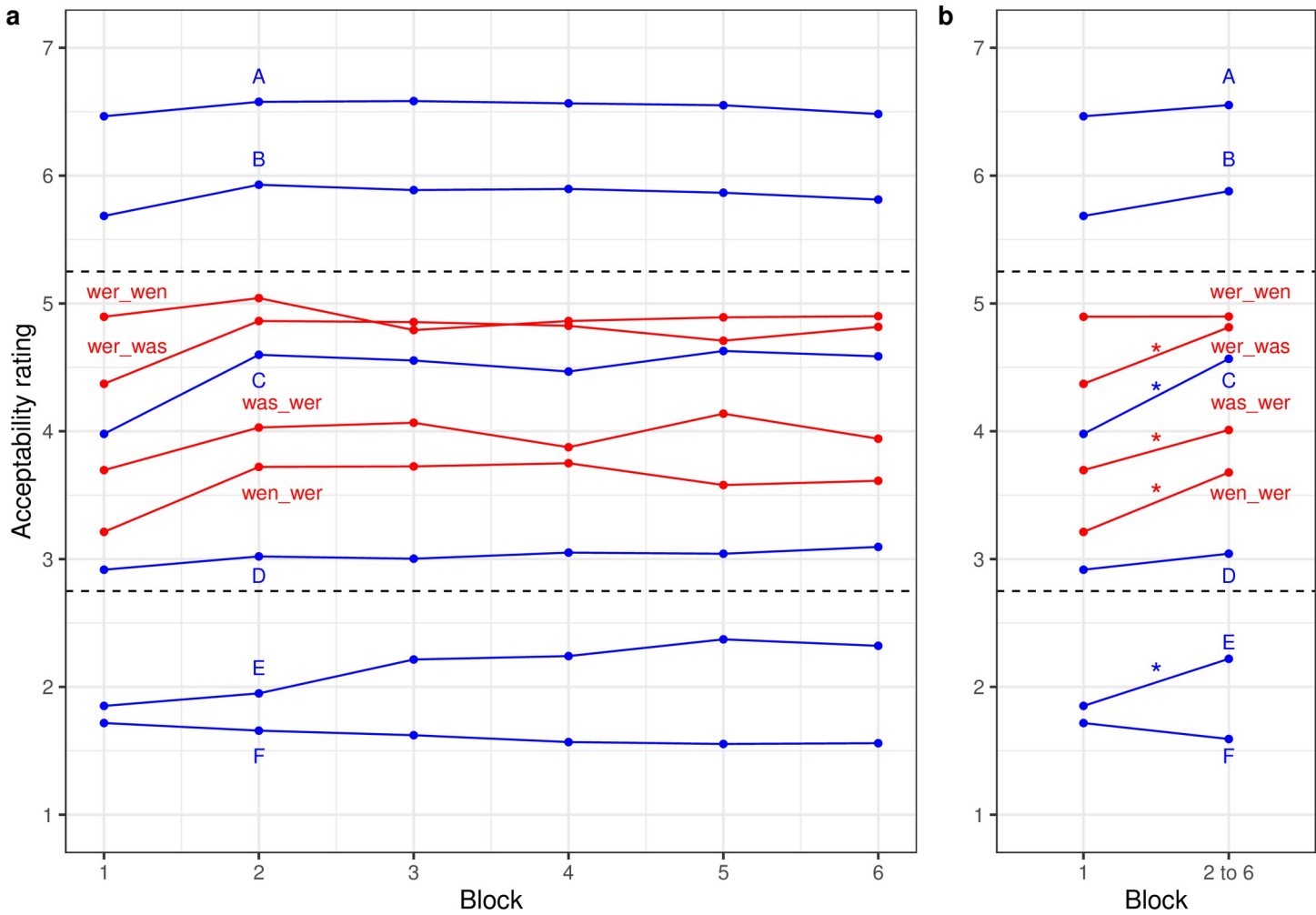

**Fig 1.** Well-formedness ratings (a) by block and (b) for first block and mean of later blocks for German targets and fillers. Dashed lines indicate the upper and lower bounds of the satiation zone based on the 25% criterion. Asterisks indicate significant changes (p<0.05).

interpretable), suggesting that the sentence is not categorically ill-formed. This finding is consistent with the claim that German has no superiority condition in the grammar and shows only selective superiority effects in cases where processing difficulty is increased (Haider [25]).

Finally, there is a general increase in well-formedness from the first to the second block of trials and this increase was stronger for "wer-was" constructions than the other three. Thus, it appears that the difference in well-formedness with an animacy mismatch in subject-initial targets can be overcome with modest exposure with the mismatch condition.

**Targets in the context of fillers.** Is the increase in well-formedness from block 1 to block 2 related to sentence type–and therefore to natural syntactic classes—or is it rather a reflection of the general level of well-formedness? We use the ratings of fillers to address this question. As with targets, there was no significant change in well-formedness ratings from block 2 to block 6 for the levels of fillers relevant for the comparison with targets. Therefore, we averaged the performance of the final five blocks for the ten types of sentences. Fig 1B shows the corresponding pattern of changes between block 1 and the mean of blocks 2 to 6.

The four types of targets were presented together with six different levels of fillers A to E which were expected to cover the spectrum from clearly grammatical (A and B) to increasingly

ungrammatical (C to F). Experience with "unusual" grammatical targets that are licensed by the grammar (i.e., object-initial targets) should lead to a larger gain in ratings of well-formedness than typical grammatical fillers like A and B (i.e., the latter should already be rated at a very high level–there is no or only little room for improvement) and also to a larger gain than for ungrammatical fillers like C to F (i.e., they should be rated at a low level and stay there because they are not licensed by the grammar).

In Fig 1B, a continuous core of changes in the 2.75 to 5.25 zone were significant (wer_was: b = 0.44, z = 3.31; C: b = 0.59, z = 4.00; was_wer: b = 0.31, z = 2.11; wen_wer: b = 0.47, z = 3.45). The highest and lowest Types within the 2.75 to 5.25 zone, namely wer_wen and filler level D, did not show significant change (|z-values| < 0.89). Counter to expectations, Filler E (b = 0.37, z = 4.43) changed significantly although it was outside of the zone. Finally, Types A, B, and F, all of them expected outside the zone, did not change significantly (all |z-values| < 1.82).

The experiment therefore fulfilled our expectations only partially. We do see pervasive satiation in middle ratings. However, first, the relevant zone appears to be more restricted than hypothesised, and, second, we also observe the rise of filler E, outside of the satiation zone; the latter occurred with a delay.

In summary, the German data shows that (i) regardless of exposure, the strength of superiority effects can be modulated by manipulating animacy (here we replicate previous findings in the literature, e.g., Häussler et al. [23]); (ii) exposure effects can be reliably induced experimentally; and (iii) exposure effects may be also be a property of intermediate judgments rather than of certain types of syntactic constructions.

## Experiment 2: Rating of English sentences

For German, many authors assume that object-initial multiple questions are indeed grammatical because the superiority condition (or the more general constraints implying it) can be circumvented or fail to apply because of peculiarities in German sentence structure. When crossing movement is less acceptable, such differences in intuitions are taken to result from grammar-external factors such as increased processing complexity (e.g., Haider [25]). Do patterns of exposure effects change in a language that has a grammatical superiority condition? In this section, we report a (partially) parallel study to the German one in English, where it is still controversial whether ungrammaticality in superiority violations is caused by the grammar or not (e.g., Hofmeister et al. [26], Häussler et al. [23] for recent discussion). We test whether we find the same or a different pattern of exposure effects to German.

The animacy variation in the German items had the purpose of having effects of crossing movement with different strength. Animacy variations seem to have no such effect in English (Häussler et al. [23]), but there is another differentiation with a similar consequence in English, namely *discourse-linking*. A discourse linked *wh*-phrase such as "which book" asks for a specific item among a contextually introduced set of objects. Discourse-linking changes the well-formedness of *wh*-island constructions, and also the well-formedness of multiple *wh*-questions by eliminating penalties for crossing movement, so that crossing movement may no longer be ungrammatical even in English (Pesetsky [42] and [43], cf. also Featherston [41] for experimental evidence supporting this assumption in the generative literature).

With these considerations, the experiment again corresponds to a 2 x 2 design with subject- or object-initial *wh*-words and with or without discourse-linked *wh*-phrases. Examples of the four English target conditions are given in (5).

 5. English target conditions

a. Condition 1: subject-initial; non-discourse-linked *wh*-phrases: who–what; well-formed in English
*The housekeeper forgot **who** had dropped **what** during the party.*

b. Condition 2: subject-initial; discourse-linked *wh*-phrases: which $N_{subj}$–which $N_{obj}$; well-formed in English
*The housekeeper forgot **which guest** had dropped **which glass** during the party.*

c. Condition 3: object-initial; non-discourse-linked *wh*-phrases: what–who; ill-formed in English
*The housekeeper forgot **what who** had dropped during the party.*

d. Condition 4: object-initial; discourse-linked *wh*-phrases: which $N_{obj}$–which $N_{subj}$; not grammatically ill-formed in English
*The housekeeper forgot **which glass which guest** had dropped during the party.*

The first factor 'superiority' can be seen by comparing (5a-b) with (5c-d). In the examples in (5a) and (5b), the subject appears before the object and thus fulfills the superiority condition. Examples (5c) and (5d) show crossing movement, and at least (5c) violates the superiority-condition in most if not on all accounts. In terms of linear order, (5d) should fall into the same category, but at least some models take the superiority condition to be inapplicable here. The second factor of discourse-linking can be seen by comparing (5a) and (5c) with (5b) and (5d). In (5a) and (5c), the subject and object are the non-discourse-linked *wh*-words *who* and *what*, whereas in (5b) and (5d) the subject and object are the discourse-linked DPs *which guest* and *which glass*.

## Method

**Subjects.**   A total of 48 English native speakers participated in the study. Fourteen self-identified as male, 32 self-identified as female and 2 subjects selected "other" under gender. Ages ranged between 18 and 48, with an average age of 31 years. Recruitment was carried out through the web-based recruitment platform, Prolific. Participation was voluntary and was remunerated with £6.80. All participants gave written consent, and were naive with regard to the questions and objectives of the study.

**Apparatus.**   The experiment was conducted using Ibex software on the web-based Ibex Farm server (https://spellout.net/ibexfarm, developed by Alex Drummond). For the lab-based study, we had generated distinct questionnaires for each participant. To retain distinct counterbalancing in this web-based version, we created 48 distinct questionnaires with unique links. Participants first clicked on a welcome page and then received a unique link, in randomised order. Clicking on the link led them to the questionnaire. The stimulus material was presented on a computer screen along with the numbers 1 to 7 arranged horizontally in small boxes. A short description of the degree of well-formedness was given under the numbers 1 "not at all well-formed" and 7 "completely well-formed". No description was given for numbers 2 to 6. Sentence ratings of "1" to "7" could either be entered on a keyboard or by pointing and clicking on the number.

**Material.**   As in the German study, material consisted of 120 targets and 252 fillers. The targets were translations of the German material with some adjustments. Instead of animacy we implemented discourse-linking as a second factor. We also left out adverbs because including adverbs in English did not make the sentences sound more natural, and changed the content nouns in some sentences. A complete list of targets and fillers is provided in the OSF Repository (https://osf.io/ge2db/).

As for German sentences, targets represent indirect multiple *wh*-questions in subordinate clauses as shown above in (5). The fillers represented six levels of well-formedness, as with German. Each level of well-formedness was made up of 42 items. The top five levels were each made up of three constructions that were identified in Gebrich et al. [11] as consistently rating at that gradient level. To create the fillers, we used the three example items from Gebrich et al. [11] for each level and created additional items until we reached the desired number of 42. We then added an additional sixth gradation (F) "not at all well-formed" to better reflect the lower end of the spectrum of acceptability.

6.   English filler examples (based on Featherston fillers)

a. Level A:

The patient fooled the dentist by pretending to be in pain.

There is a statue in the middle of the square.

The winter is very harsh in the North.

b. Level B:

Before every lesson the teacher must prepare their materials.

Jane does not boast about her being elected president.

Jane cleaned her motorbike with which cleaning cloth?

c. Level C:

Hannah hates but Linda loves eating popcorn in the cinema.

Most people like very much a cup of tea in the morning.

The striker must have fouled deliberately the goalkeeper.

d. Level D:

Who did she whisper that had unfairly condemned the prisoner?

The old fisherman took her pipe out of mouth and began story.

Which professor did you claim that the student really admires her?

e. Level E:

Historians wondering what cause is disappear civilisation.

Old man he work garden grow many flower and vegetable.

Student must read much book for they become clever.

f. Level F:

The ink was for spilled.

**Design and counterbalancing.**   Counterbalancing was identical to the German study. In order to implement the individualised counterbalancing scheme over the internet, we created unique questionnaires for each participant ID, and assigned participants to IDs using a random link generator in Ibex Farm.

**Procedure.** Subjects were instructed to judge the well-formedness of the sentences according to their judgments of spoken language, rather than written language that they might find in a textbook, and to use the full spectrum of the seven-point scale. They practiced the procedure with ten examples. Then, each subject worked through the six blocks of sentences; durations of breaks between blocks was under the subjects' control.

**Statistical analysis.** Statistical analysis for the English study followed the same procedure as in Experiment 1. Details of model selection are documented in the analysis scripts in the OSF repository. This procedure was identical to the one used for experiment 1 and led to a very similar random-effects structure. Estimates of model parameters and comparative goodness of fit statistics are reported in the supplement.

## Results and discussion

**Targets.** The four types of targets using constructions of (a)"who-what", (b) "which $N_{subj}$-which $N_{obj}$", (c) "what-who", and (d) "which $N_{obj}$-which $N_{subj}$" for the same sentence frames map onto a 2 x 2 design with the main effects subject-vs-object initial targets (so; a+b-c-d) and discourse-linking of subject and object (dlink; a+c-b-d) as well as the interaction between these two effects (a-b-c+d). In addition, this 2 x 2 design was repeated across six blocks of trials. Fig 2A displays the change in ratings of well-formedness for the four types of targets shown in red; Fig 2A also shows performance for the levels of fillers. The LMM fixed-effect estimates are provided in the supplement.

Here again, the two subject-initial targets were rated as more acceptable than the two object-initial ones, yielding a significant main effect of order (b = 0.67, z = 12.25). Overall, there was also a significant difference between targets with discourse-linked *wh*-phrases to targets without discourse-linked *wh*-phrases (b = -0.35, z = -6.76) qualified by a significant interaction with order (b = 0.43, z = 9.49): Subject-initial word order was clearly preferred with or without discourse-linking to the other two conditions; for object-initial word order there was a clear preference for the discourse-linked construction ("which $N_{obj}$−which $N_{subj}$").

The change in ratings across blocks showed an overall profile that was less clear than for German targets: There was a nominal, but not significant increase from block 1 to the average of the following blocks (b = 0.16, z = 1.77) and, counter to expectations, a significant overall negative difference between block 3 and the final three blocks (b = -0.11, z = -2.47). However, there were two significant interactions between the first and second contrast for block with order. There was a significant increase in well-formedness from the first block to the average of the others for object-initial but not for subject-initial targets (b = -0.29, z = -4.95) and a weaker effect for the change from the second to the average of the rest (b = -.14, z = -.3.17). The second interaction was "helped" to some degree by a small reduction of well-formedness of subject-initial targets in the final blocks. Finally, there was also a significant interaction for the second contrast of block with discourse-linking due to an increase in well-formedness for the non-discoursed linked targets ("what-who"; b = 0.10, z = 2.28). These are the targets with the lowest well-formedness of the four conditions.

**Targets in the context of fillers.** As with German sentences, it is instructive to examine which sentence types exhibited a change across blocks, especially from the first to the second one (see Fig 2). In Fig 2B, all changes in the 2.75 to 5.25 zone were significant (OS_which: b = 0.32, z = 2.52; C: b = 0.46, z = 3.96; OS_wh: b = 0.58, z = 4.46; D: b = 0.42, z = 3.61). Moreover, with one exception, none of the Types outside the zone were significant (all |z-values| < 1.35). The exception, as for German sentences, was the significant rise of Filler E (b = 0.40, z = 4.72).

Thus, the English sentences show pervasive satiation in the 2.75 to 5.25 zone of judgment, as expected. Unlike for German sentences, there is no ambiguity in this respect even for fillers of level D.

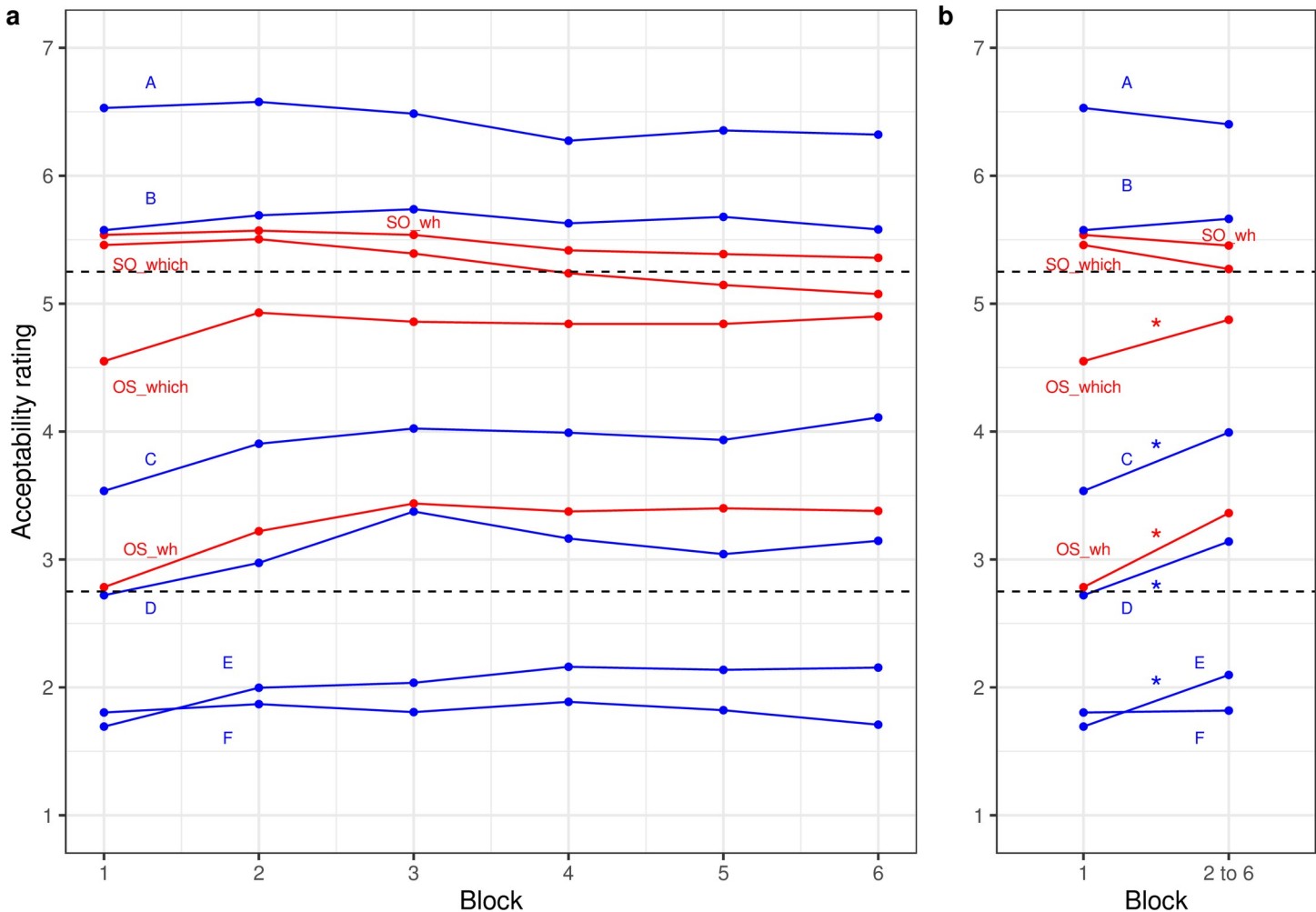

**Fig 2.** Well-formedness ratings (a) by block and (b) for first block and mean of later blocks for English targets and fillers (web experiment). Dashed lines indicate the upper and lower bounds of the satiation zone based on the 25% criterion. Asterisks indicate significant changes (p<0.05).

In summary, in English targets, just like in the German experiment, there is a large difference in the judgement of well-formedness of multiple *wh*-questions with a clear preference for subject-initial over object-initial targets, in line with previous experiments (Sprouse [44], Hofmeister et al. [26]). For object-initial targets there was a very clear preference for discourse-linked than not-discourse-linked targets. There was positive initial change for the well-formedness of object-initial targets, but this change might simply reflect a general middle-raise in well-formedness also observed for fillers with a similar initial rating of well-formedness.

## Experiment 3: Rating of English sentences (replication in lab)

In Experiment 3, we report a replication of the English experiment. Time-wise, this experiment was carried out after Experiment 1 and before the web-based Experiment 2. We chose to run Experiment 2 because the counterbalancing scheme was not rendered as intended in Experiment 3. Although the targets were not counterbalanced as intended across blocks in Experiment 3, some aspects of the counterbalancing were preserved: each participant still saw an individual questionnaire, as well as more than one target item per block. Fillers were counterbalanced across blocks as intended, and ordering restrictions such as making sure that

targets were not adjacent were maintained. We submit that past studies counterbalanced target conditions, but, as far as we could tell from procedural descriptions, earlier studies did not counterbalance conditions across blocks either.

Most importantly, one important result of Experiments 1 and 2 was that the change in well-formedness between initial blocks of trials for targets might simply reflect their level of well-formedness. This argument was based on the absence of evidence for differences in change when compared to ungrammatical sentences with comparable ratings of well-formedness. The argument rests on arguing a null hypothesis of parallel changes. Such results are in need of replication.

## Method

**Subjects.**    A total of 48 subjects (28 female, 20 male) participated in the study. Ages ranged between 16 and 51, with an average age of 23 years. Recruitment was carried out through the participant pool (SONA) at University College London (Psychology and Language Sciences) and the University of Cambridge (Language Sciences). During data analysis it became clear that counter to expectation there was one 16-year old participant included for Experiment 3. There should not have been 16-year olds in the pools we recruited from. The record may be a typographical error or it may be a very young university student. We cannot find out because we cannot link the record to the participant. Participation was voluntary and was remunerated with £6. All participants gave written consent, and were naive with regard to the questions and objectives of the study.

**Apparatus.**    The experiment was conducted in the Speech and Language Sciences Lab at University College London and at Trinity Hall, Cambridge. The experiment was implemented using the Python software PsychoPy [30] version 1.84.2. The stimulus material was presented on a computer screen along with a scale labelled from 1 to 7. Underneath each of the numbers was a short description of the degree of well-formedness: 1 "not at all grammatically natural", 2 "almost not grammatically natural", 3 "more grammatically unnatural than natural", 4 "cannot be rated", 5 "more grammatically natural, than unnatural", 6 "almost grammatically natural", 7 "completely grammatically natural". Sentence ratings from "1" to "7" were entered on a standard computer keyboard with a German keyboard layout.

**Material.**    Material was identical to Experiment 2.

**Design and counterbalancing.**    We did not counterbalance the targets across blocks in this study, due to an error in generating the questionnaires. Some of the features of the counterbalancing scheme used in Experiment 1 and Experiment 2 were nonetheless retained, such as different questionnaires for each participant, and counterbalancing of fillers across blocks.

**Procedure and statistical analysis.**    Subjects were instructed to judge the well-formedness of the sentences according to their judgments of spoken language, rather than written language that they might find in a textbook, and to use the full spectrum of the seven-point scale. They practiced the procedure with ten examples. Then, each subject worked through the six blocks of sentences; durations of breaks between blocks was under the subjects' control. Statistical analysis for Experiment 3 followed the procedure for the first two experiments. Details of model selection are documented in the analysis scripts in the OSF repository. Again, the procedure was identical to the one used for Experiments 1 and 2 and led to a very similar random-effects structure. Estimates of model parameters and comparative goodness of fit statistics are reported in the supplement.

## Results and discussion

**Targets.**    Fig 3A displays the change in ratings of well-formedness for the four types of English targets shown in red and the six types of fillers in blue. The LMM fixed-effect estimates

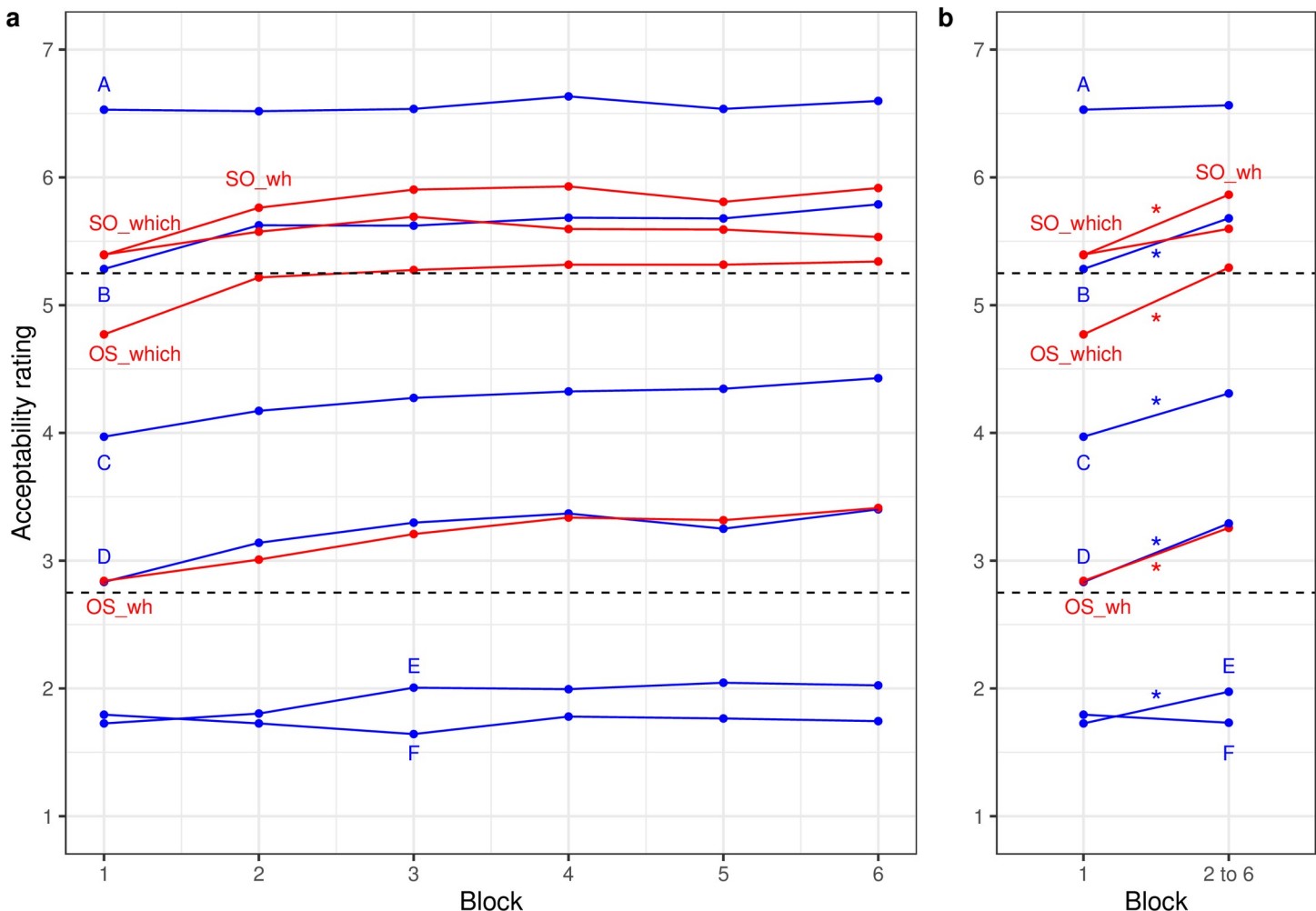

**Fig 3.** Well-formedness ratings (a) by block and (b) for first block and mean of later blocks for English targets and fillers (lab experiment). Dashed lines indicate the upper and lower bounds of the satiation zone based on the 25% criterion. Asterisks indicate significant changes (p<0.05).

and statistics are provided in the supplement. Main effects of order (b = .74, z = 12.93) and discourse-linking (b = -0.45, z = -8.87) as well as the interaction of these two factors (b = 0.56, z = 10.97) were replicated in the lab experiment: again, subject-initial word order was clearly preferred with or without discourse-linking to the other two conditions; for object-initial word order there was a clear preference for the discourse-linked construction ("which $N_{obj}$–which $N_{subj}$").

The change in ratings across blocks was much clearer than in Experiment 2: The first two block contrasts were significant (b1: b = 0.39, z = 4.54; b2: b = 0.14, z = 3.53) indicating an overall increase of well-formedness from the first to the average of the rest and a second, smaller increase from block 2 to the rest. Again, there were two significant interactions between the first and second contrast for block with order. There was a significant increase in well-formedness from the first block to the average of the others for object-initial but not for subject-initial targets (b = -0.13, z = -2.24) and for the change from the second to the average of the rest (b = -.12, z = -.3.09). As in Experiment 2, the increase in well-formedness across the initial blocks was larger for object-initial than subject-initial targets (see Fig 3B). Unlike in Experiment 2 there was also an increase for subject-initial targets.

**Targets in the context of fillers.** The final question again is which sentence types changed across the two initial blocks. And again, as shown in Fig 3B, all Types in the 2.75 to 5.25 zone rose significantly (OS_which: b = 0.57, z = 3.76; C: b = 0.34, z = 2.39; D: b = 0.42, z = 4.60; OS_wh: b = 0.44; z = 3.70). In addition, with initial values between 5.25 and 5.30 just barely outside the upper boundary of the zone, there was also significant rise for one target (SO_wh: b = 34, z = 2.42) and one filler (B: b = .40, z = 3.16) Type. Neither SO_which (outside the zone) nor top and bottom fillers A and F, both clearly outside the zone, changed (all |z-values| < 1.35). Finally, as in the preceding two experiments, filler E rose significantly despite being outside of the 2.75 to 5.25 zone of satiation (b = 0.25, z = 3.62).

## General discussion

### Summary of combined results

We found parallel patterns between German and English for both exposure effects and modulations of their size by crossing movement and by factors related to semantics and discourse. Specifically, we found:

a. A rise in initial block(s) only;

b. A rise in a continuous zone of acceptability irrespective of sentence type, that is shifts were found (i) in target conditions that respected superiority; (ii) target conditions that violated superiority; and (iii) filler levels with no *wh*-elements at all;

c. Comparable shifts in German and English, both in initial blocks and in continuous zones of well-formedness;

d. Effects of D-linking in English that were on par with animacy effects in German, specifically:

 i. A decrease for German with superiority violations with matching animacy that was comparable to regular (non-d-linked) superiority violations in English

 ii. An increase for English with superiority violations where both *wh*-phrases were D-linked that was comparable to acceptable (mismatched animacy) superiority violations in German.

### Satiation effects track zone of well-formedness, not grammatical construction

Our experiments sought to establish whether there are "satiation effects" in the sense of studies building on Snyder [5] caused by repeated exposure, and whether these effects are zone- or construction-sensitive. All three experiments showed a clear rise in acceptability for some of the conditions of the target experiment and for some of the filler groups between the first and the remainder of the blocks. It is difficult to conceive of an alternative account for these effects observed across three experiments in two languages than the assumption that ratings increased with the number of items perceived in a certain target experimental condition or in a certain filler group.

Is satiability driven by syntactic properties or does it indiscriminately affect all constructions of a certain zone of acceptability? The results of our experiments by and large support the latter hypothesis. Both in the German and the two English experiments, there was a continuous zone (with two exceptions discussed below) in the range of acceptability such that all constructions in that zone satiated, and no items outside that zone showed an increase in

acceptability. This was the case even though the constructions affected have no obvious grammatical factor in common. In German, for instance, ratings rise for the two object-initial multiple questions (target conditions 3 and 4), for the well-formed subject-initial questions with mismatching animacy (target condition 2), and for fillers C which involve unusual binding options and ill-formed orders of dative and accusative objects. One obvious factor that these constructions do have in common is their level of acceptability. In English we see the same rise in acceptability across dissimilar constructions as in German, despite the effects pertaining to "harder" grammatical constraints. For instance, crossing movement improves, but so do constructions that do not respect verb-object adjacency constraints, constructions where the subject pronoun *who* is ungrammatically extracted across the complementiser *that*, and where pronouns do not match the gender of their antecedent, for example, *fisherman—her*. What unifies all these constructions is that participants give them ratings in the same range.

Our results come with two results that are inconsistent with the zone argument. First, filler D did not satiate in German. This could merely indicate that the satiation zone of Experiment 1 begins at quite a high point in the judgment space (the result would then be less of a problem) or it could point to the existence of a set of constructions that are well in the satiation zone. Filler D did show satiation with the English constructions, meaning that this problem was restricted to the German language or German materials.

The rise of out-of-zone filler E in all three experiments is an interesting, but special case in relation to filler F which we had intended to represent an extreme ungrammaticality, much more severe than the one represented by filler E. Counter to expectations, subjects judged both of them equally ungrammatical during the first block. However, with some exposure they did recognize the difference in severity of ungrammaticality. Therefore, it seems that the increase in acceptability does not really reflect a change in acceptability, but rather the recognition of the more severe type of ungrammaticality that we had intended to show up from the start.

Apart from the two difficulties just discussed, our conclusion concerning the postulation of a "zone of satiation" can be criticised in at least two ways. With respect to the current experiments, it might be objected that the satiating constructions do have a common grammatical or psycholinguistic property overlooked by the present authors, or that the factors making satiation possible are manifold, such that all our relevant items simply happen to fall under one or the other of these factors. Both possibilities are real, but difficult to refute in the absence of a concrete proposal as to what these factors might be.

## Dynamics of satiation

Our results go against the expectation that satiation effects rise continuously to an asymptotic level of maximal well-formedness. There is an initial early increase followed by stability clearly below the maximum. This result may well be the most remarkable feature of the plots of judgments across blocks and across the three experiments. The size of initial rise of ratings is not very dramatic either; they increase only by roughly .5 on a 7-point scale, meaning that repeated exposure does not change the quality of the judgment. The relatively small size of the effect is in agreement with most of the experimental results reported in the literature. The stability we found could be (1) due to predominantly stable judgments, (2) due to participants developing a rating pattern in the first two blocks that is used for the remainder of the experiment (entrenchment), or (3) due to a "mere exposure effect" (Zajonc, [45]).

If the first option applies, participants might have stopped deliberately rating the item they are presented with and instead produced memorised values to assign to types of sentences. There is currently no accepted way to control for such memorising and entrenchment. Some limited discussion in previous works like Sprouse [13] presents a potential argument that

memorising and entrenchment undermine a balanced design and the use of sets of items of the same type, as is the case with control items and fillers. Alternatively, according to the second option, participants might have taken their time to find a stable mapping from their perceived intuitions to a 7-point scale. Once that mapping is established ratings remain constant. Both explanations, however, do not explain why the stabilization of the judgments is always upwards-oriented.

The "mere exposure effect" refers to observations that experimentally manipulated frequency of exposure to nonsense words and syllables increases their ratings of positive affective connotations (e.g., familiarity, liking, etc.) without any reinforcement (Zajonc, [45]). Bader and Häussler [46] obtained such a correlation for printed frequency of sentences and their rating of well-formedness. We envision that this account could open a promising line of research, especially if consequences for ratings of acceptability due to differences in the quality of prosodic representations between sentences varying in degree of well-formedness are considered as well.

## The integrated analysis of targets and fillers

We end with a methodological note. Is it appropriate to compare within-item targets to between-item fillers? A norm in the field of experimental syntax is to test only within-item differences (Sprouse et al. [47]). Within-item factorial designs are indeed a powerful way to control for item variability. In a principled theory-driven workflow, within-item designs allow us to test theory-relevant contrasts specified *a priori* with tight control of known sources of the variability. Fillers, however, should be as unrelated as possible to the targets. Deriving them as within-item variants of targets. However, the fillers used in these experiments are not just any fillers that we averaged into groups after testing. Rather, we had *a priori* expectations about the gradient acceptability of these fillers based on the norming studies in Featherston [10] and Gebrich et al. [11]. We also carefully built the additional fillers that we used to the template (*Eichsatz*) used in the norming study. Moreover, results did indeed meet expectations that we had before testing; they differed significantly in the expected order. Therefore, although part of the analysis presented in this paper is exploratory–in particular, we decided which target conditions to compare to which other target or filler conditions after seeing the results–we chose the levels of acceptability before testing based on previous research results. It may turn out that future research will uncover interesting commonalities between fillers that we have missed here. Such a development is part and parcel of programmatic research however, and does not undermine the validity of comparing targets to normed levels of gradient fillers, provided that norming is carried out before testing and the differences between acceptability levels come out as expected.

## Conclusion

In conclusion, this paper revisited old questions about satiation, superiority effects and processing complexity of exposure effects across a single experimental session. The combination of (1) targeting superiority violations, (2) integrating six blocks of trials as an experimental factor into the counterbalancing scheme, (3) going beyond previous research with respect to number of subjects and number of items, (4) integrating carefully selected levels of filler sentences in a secondary joint analysis of targets and fillers, (5) replicating the overall profile of means in two languages, that is German (Experiment 1) and English (Experiments 2 and 3), and (6) replicating, for English, the overall profile of means between web (Experiment 2) and lab (Experiment 3) should provide a useful reference platform for follow-up research. Moreover, all our data and scripts are available for additional exploratory re-analyses from different

theoretical perspectives. We neither claim that we resolved all open questions–indeed there are a few results that are inconsistent with our perspective–nor that the results generalise far beyond the experimental setting (e.g., results may change with comprehension questions, see Zervakis & Mazuka, [9]).

The main results are that after an initial rise, there was a remarkable quantitative asymptotic stability of acceptability ratings within experiments and there was also remarkable qualitative agreement between experiments with German and English sentences for strong differences in syntactic violations. The results corroborated claims of satiation effects and strongly suggest that the nature of these effects is different than claimed in previous studies: rather than a continuous rise to maximal acceptability, judgments rise only initially and asymptote significantly below the maximum. Thus, the effects appear to be primarily linked to an intermediate zone of well-formedness, not to natural syntactic classes.

## Supporting information

**S1 Table. Experiment 1 (lab, German): LMM fixed-effect estimates for target ratings.**
(DOCX)

**S2 Table. Experiment 1 (lab, German): LMM fixed-effect estimates for block effect within sentences types.**
(DOCX)

**S3 Table. Experiment 2 (web, English): LMM fixed-effect estimates for target ratings.**
(DOCX)

**S4 Table. Experiment 2 (web, English): LMM fixed-effect estimates for block effect within sentences types.**
(DOCX)

**S5 Table. Experiment 3 (lab, English): LMM fixed-effect estimates for target ratings.**
(DOCX)

**S6 Table. Experiment 3 (lab, English): LMM fixed-effect estimates for block effect within sentences types.**
(DOCX)

**S7 Table. Experiment 3 –English sentences (lab): Secondary LMM goodness of fit statistics and parameter estimates.**
(DOCX)

## Acknowledgments

We would like to thank the four reviewers for their very helpful feedback and suggestions. We would also like to thank Kim-Laura Speck for help developing the counterbalancing scheme used in the experiments, and Johannes Rothert for extracting the values in Table 1 from the relevant publications.

## Author Contributions

**Conceptualization:** J. M. M. Brown, Gisbert Fanselow, Rebecca Hall, Reinhold Kliegl.

**Data curation:** J. M. M. Brown, Rebecca Hall, Reinhold Kliegl.

**Funding acquisition:** Gisbert Fanselow, Reinhold Kliegl.

**Investigation:** J. M. M. Brown, Rebecca Hall.

**Methodology:** J. M. M. Brown, Gisbert Fanselow, Rebecca Hall, Reinhold Kliegl.

**Resources:** Reinhold Kliegl.

**Software:** J. M. M. Brown, Reinhold Kliegl.

**Supervision:** Gisbert Fanselow, Reinhold Kliegl.

**Writing – original draft:** J. M. M. Brown, Reinhold Kliegl.

**Writing – review & editing:** J. M. M. Brown, Gisbert Fanselow, Reinhold Kliegl.

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
