## [Decision Letter · Decision Letter 0]

24 Nov 2020

PONE-D-20-32419

Middle ratings rise regardless of grammatical construction: Testing syntactic variability in a new repeated exposure paradigm

PLOS ONE

Dear Dr. Brown,

Thank you for submitting your manuscript to PLOS ONE. After careful consideration, we feel that it has merit but does not fully meet PLOS ONE’s publication criteria as it currently stands. Therefore, we invite you to submit a revised version of the manuscript that addresses the points raised during the review process.

In particular, it is necessary to explain in detail why it is important to use the block design and the superiority effect for your study, in addition to accommodate all the other issues, both major and minor, raised by the reviewers.

We look forward to receiving your revised manuscript.

Kind regards,

Masatoshi Koizumi, Ph.D.

Academic Editor

PLOS ONE

Journal Requirements:

3. Please remove your figures from within your manuscript file, leaving only the individual TIFF/EPS image files, uploaded separately.  These will be automatically included in the reviewers’ PDF.

Reviewers' comments:

Reviewer's Responses to Questions

**Comments to the Author**

1. Is the manuscript technically sound, and do the data support the conclusions?

Reviewer #1: Partly

Reviewer #2: Yes

Reviewer #3: Yes

Reviewer #4: Partly

2. Has the statistical analysis been performed appropriately and rigorously? 

Reviewer #1: Yes

Reviewer #2: I Don't Know

Reviewer #3: Yes

Reviewer #4: Yes

3. Have the authors made all data underlying the findings in their manuscript fully available?

Reviewer #1: Yes

Reviewer #2: Yes

Reviewer #3: Yes

Reviewer #4: Yes

4. Is the manuscript presented in an intelligible fashion and written in standard English?

Reviewer #1: Yes

Reviewer #2: Yes

Reviewer #3: No

Reviewer #4: No

5. Review Comments to the Author

Reviewer #1: The study reports that ratings rise with repeated exposure at the beginning of the experiment regardless of syntactic construction. The ms claims that rather than a continuous rise, judgments rise only initially, which in a sense is a null effect: absence of satiation throughout the experiment.

I am not a native speaker of German and so I can't try to understand why there was little improvement in some of these constructions. However, it seems to me very likely that the initial rise in acceptability may be simpyly due to speakers adjusting to the task and items, once the actual experiment begins. After a few items, informants may adjust the way they are using the scale.

But the biggest problem is that there seem to have been no comprehension questions, and given the size of the experiment (120 target items + and 252 filler sentences), which is categorically excessive. If I had to read 372 weird (= 'out of the blue') sentences and give them some number for a small chunk of money, my motivation to do it carefully would be low and fatigue would definitely play a role on my attention. Indeed, tt is perfectly possible that speakers did not pay too much attention to the items, and gave good-enough ratings, which ended up cancelling each other for the most part. Hence, there is a rise at the beginning of the experiment but once fatigue and boredom sets in, there is no motivation to maintain a high level of attention.

There are also various confusing claims in this ms, that need clarification. I am unclear on why the block design is claimed to be superior to a linear design. If the claim is that acceptability changes with repeated exposure, then the most natural experimental set-up is one where each exposure is its own bin. The [19] and [24] studies can be seen as block designs in which there are as many blocks as experimental items, since each presentation position corresponds to different sentences (in the same condition) per informant. One can fit a loess line on the results. In contrast, the block design is a coarser-grained design, and therefore, not ideal. I really must be missing something...

Furthermore, one can argue that the number of blocks that was chosen is not appropriate to measure all possible satiation dynamics. Maybe the blocks are sometimes too wide, and sometimes too narrow, to capture shifts in acceptability change. Everyone implicitly assumes the effect of satiation is approximately linear, but there is absolutely no reason for this to be the case. I've seen satiation results in which there are non-linear bumps in the middle of the experiment and experiments where the acceptability dips in the last items (there are various reasons why this can happen, but the point is that it can happen). I am not convinced this type of design is unproblematic and superior to others.

The ms goes on to state that the studies [19] and [24] are merely testing ordering effects, not satiation effects per se. I find this puzzling. In [19] and [24] no two participants saw the items in the same order, which means that the results cannot be an ordering effect. Again, I must be missing something.

"Third, these effects are consistent across languages" should be reworded more conservatively as "Third, these effects are consistent across these two languages, for the items tested".

Reviewer #2: This is a careful and thoughtfully done study with many interesting conclusions, though ultimately it may raise more questions than it answers (which is not necessarily a bad thing). Unlike many experiments testing syntactic satiation, this one presents a very large number of stimuli for each target structure, uses a rigorous block design and careful counterbalancing, and is done in virtually identical format across two languages.

The paper could be greatly improved if certain crucial aspects were explained better:

• Given the important role that blocks play in the experimental design, it is surprising how little explanation is given as to why these are important and what is innovative about the block design in the current experiment. An attentive reader can piece things together, but the reader really shouldn’t have to do this.

• The empirical focus of the satiation experiment is the superiority effect, but very little explanation is given as to why this phenomenon was chosen, beyond the statement that it has “a high degree of variability” (but “variability” in what sense?). Since one of the most interesting claims in the literature is that only certain structures will exhibit satiation, the question of why this structure in particular was chosen deserves more discussion.

• The discussion of the materials for the experiment is especially hard to figure out. Partly, this is because the word “sentence” is used sometimes to refer to an actual sentence and sometimes to a set of sentences (what is sometimes called in other papers a “lexicalization set” or “token set”), which means the reader often has to guess what is intended. The numbers of stimuli are also hard to figure out. At one point, it is stated that each block consists of 48 (4 x 12) experimental items and 24 filler items, but then somehow the total number of items in a block is 62. Getting these details right is important, because the reader needs them understand what the experimental-filler ratio is. It is not separately given, but there would appear to very few fillers relative to the experimentals. Once again, though, the reader should not have to be the one figuring this out. The ratio used should be explicitly stated and justified.

• As mentioned, blocks are a crucial design feature of this experiment, but it is not made clear what role they play while stimuli are presented to participants. The paper states that participants may rest for as long as they want between blocks, but it isn’t clear how they know when each block finishes (given that in other studies, divisions between blocks are invisible to participants). It also isn’t stated how long the experiment took, which is potentially important, since the number of stimuli is significantly longer than standard acceptability experiments, which raises possible questions of fatigue, attention to task, etc.

• At the very end of the paper, the distinction between strong and weak islands is brought up for the first time without any context. The authors seem to suggest that their results have something to say about this distinction, but it isn’t clear what it is.

• At several points in the paper, the expression “as for” is used when it appears that “as with” was what was actually intended (e.g., “as for German” when the context makes “as with German” more plausible).

Reviewer #3: Review Comments to the Author

Please use the space provided to explain your answers to the questions above. You may also include additional comments for the author, including concerns about dual publication, research ethics, or publication ethics. (Please upload your review as an attachment if it exceeds 20,000 characters) (Limit 200 to 20000 Characters)

Could you please see my attached review, I have added all my comments to the Author in an attachment.

Reviewer #4: The study is very interesting and important as it informs an ongoing debate about language processing, parsing, and grammaticality vs. acceptability. I also agree that it would be nice to have a proper standardized experimental design to test for effects of repeated measures (the literature is mixed and some use a binary scale some a continuous one). However, the paper could do with a serious revision and it has a large number of unresolved issues. For example, it is not explained what the zone of well-formedness, and it is not clear how the results go against what is actually argued in the literature. Also, some arguments are based on comparisons between controlled target sentences and mean values for conflated sets of uncontrolled fillers; how this licensed needs to be explained. Some sections are almost impossible to understand, and there are many unexplained terms.

I recommend major revision.

6. PLOS authors have the option to publish the peer review history of their article (what does this mean?). If published, this will include your full peer review and any attached files.

Reviewer #1: No

Reviewer #2: **Yes: **Grant Goodall

Reviewer #3: No

Reviewer #4: **Yes: **Ken Ramshøj Christensen

---

## [Author Response · Author response to Decision Letter 0]

15 Feb 2021

We are grateful for the many constructive criticisms, questions, and corrections in the four reviews. Addressing them led to major changes that we think considerably improved the manuscript. We hope that the revision and the following responses to the specific issues address all comments in a satisfactory way. We have made some major changes to the manuscript, following suggestions.

The major changes are the following. We reorganized and rewrote large parts of the introduction. Specifically, we focus on the core message, namely that satiation is closely related to the intermediate nature of acceptability. We use the literature review to highlight the diversity of approaches, both in terms of substantive topics and methodologies, yielding a somewhat incoherent and sometimes contradictory profile of results in this field of research. Against this background, we provide the reasons motivating our use of superiority violations in the experiment and the role of the fillers in this context. We also revised the description of our methodological approach. We also thoroughly revised the general discussion by focusing on the evaluation of the experiments related to the guiding question.

Overall we hope that it is clearer now why our combination of (1) targeting superiority violations, (2) integrating six blocks of trials as an experimental factor into the counterbalancing scheme, (3) going beyond previous research with respect to number of subjects and number of items, (4) integrating carefully selected levels of filler sentences in a secondary joint analysis of targets and fillers, (5) replicating the overall profile of means in two languages, that is German (Experiment 1) and English (Experiments 2 and 3), and (6) replicating the overall profile of means for English between web (Experiment 2) and lab (Experiment 3) represents an important contribution to the field of research.

We neither claim that we resolved all open questions – indeed there are a few results that are inconsistent with our own theoretical perspective – nor that the results generalize far beyond the experimental setting (e.g., when carried out with comprehension questions). We do think, however, that our results provide a very useful reference platform for follow-up research. Moreover, all our data and scripts are available for additional exploratory re-analyses from different theoretical perspectives. 

REVIEWER 1

Response: Thank you for many constructive comments.

- Reviewer 1: .. it seems to me very likely that the initial rise in acceptability may be simply due to speakers adjusting to the task and items, once the actual experiment begins. After a few items, informants may adjust the way they are using the scale.

Response: We do not disagree that this is one interpretation of the results. However, it does not explain why adjustments are always positive. In the revision we discuss the general “mere exposure effect” (Zajonk, 1968) as a mechanism that would cause this trend (p. 40, Dynamics of satiation). 

- Reviewer 1: But the biggest problem is that there seem to have been no comprehension questions, and given the size of the experiment (120 target items + and 252 filler sentences), which is categorically excessive. If I had to read 372 weird (= 'out of the blue') sentences and give them some number for a small chunk of money, my motivation to do it carefully would be low and fatigue would definitely play a role on my attention. Indeed, it is perfectly possible that speakers did not pay too much attention to the items, and gave good-enough ratings, which ended up cancelling each other for the most part. Hence, there is a rise at the beginning of the experiment but once fatigue and boredom sets in, there is no motivation to maintain a high level of attention.

Response: We agree with much of this comment. Zervakis and Mazuka (2013) repeated their rating-only experiment with one including comprehension questions. The effects disappeared or were greatly reduced. We do not claim that our results generalize to the comprehension condition. We still consider our highly reliable results, replicated across three experiments in two languages, an important contribution to an understanding of acceptability ratings in a standardized laboratory setting. This and the other proposals implicit in this comment define a research program. One paper cannot be expected to resolve all these issues, but we document them as limitations in the Conclusion of the General Discussion (p. 42).

- Reviewer 1: There are also various confusing claims in this ms, that need clarification. I am unclear on why the block design is claimed to be superior to a linear design. If the claim is that acceptability changes with repeated exposure, then the most natural experimental set-up is one where each exposure is its own bin. The [19] and [24] studies can be seen as block designs in which there are as many blocks as experimental items, since each presentation position corresponds to different sentences (in the same condition) per informant. One can fit a loess line on the therefore, not ideal. I really must be missing something..

 Furthermore, one can argue that the number of blocks that was chosen is not appropriate to measure all possible satiation dynamics. Maybe the blocks are sometimes too wide, and sometimes too narrow, to capture shifts in acceptability change. Everyone implicitly assumes the effect of satiation is approximately linear, but there is absolutely no reason for this to be the case. I've seen satiation results in which there are non-linear bumps in the middle of the experiment and experiments where the acceptability dips in the last items (there are various reasons why this can happen, but the point is that it can happen). I am not convinced this type of design is unproblematic and superior to others.

- Response: There are many ways one can analyze these data. Here is the rationale that guided the design of the experiment and inferential statistics. Given the goal to estimate satiation effects, block [of trials] has the same theoretical status as the syntactic manipulations (subject/object initial; animacy congruency of wh-word). Therefore, we specified an a priori set of contrasts for the blocks of the experimental design. Our hypothesis was that ratings would increase up to an asymptotic level, but that the block at which the asymptotic level is reached might differ between syntactic conditions; we did not expect any bumps. Given this hypothesis, Helmert contrasts are ideally suited because they are orthogonal and inform us about the block in which there is no more reliable evidence for further change. (Of course, the cost of this specification is that we might miss a downward trend.)

 Each experimental item exists in four versions (representing the 2 x 2 within item factor design). Including Block with the same number of items in each block as a design factor into the counterbalancing scheme ensured that (1) in each block every subject rates the same number items in each of the four conditions, (2) each subject sees an item in only one of its four versions, (3) each of the four versions of every item is presented equally often (when aggregated across subjects), and (4) appears equally often in each of the six blocks. Thus, block is a factor with six levels that is orthogonally crossed with the two syntactic factors. Counterbalancing eliminates any correlations between blocks and conditions and thereby maximizes statistical power for the detection of interactions. The design requires a multiple of 24 subjects. We expanded on the rationale for and description of including block of trials in the counterbalancing scheme in Introduction (pp. 15-16) and Methods (p. 19) of Exp 1).

 The alternatives specified in the comment are plausible, too, but usually (not necessarily) they are aligned with analyses of the exploratory type. Given our a priori design, we prefer to stay with the analyses as reported in the original submission. We carried out some analyses of the kind suggested in a post-hoc fashion but did not see any results that would force us to qualify our report. Moreover, our data are available online and we welcome their use by colleagues to test alternative hypotheses, presumably in an exploratory fashion. 

- Reviewer 1: The ms goes on to state that the studies [19] and [24] are merely testing ordering effects, not satiation effects per se. I find this puzzling. In [19] and [24] no two participants saw the items in the same order, which means that the results cannot be an ordering effect. Again, I must be missing something.

Response: This sentence has been removed from the ms.

- Reviewer 1: [Abstract]: "Third, these effects are consistent across languages" should be reworded more conservatively as "Third, these effects are consistent across these two languages, for the items tested". 

Response: Changed to "Third, these effects are consistent across judgements of superiority effects in English and German." We included Item as random, not fixed factors. Therefore, we claim generalizability beyond the specific items. We do not generalize to other syntactic constructions.

REVIEWER 2

- Reviewer 2: This is a careful and thoughtfully done study with many interesting conclusions, though ultimately it may raise more question than it answers (which is not necessarily a bad thing). Unlike many experiments testing syntactic satiation, this one presents a very large number of stimuli for each target structure, uses a rigorous bloc design and careful counterbalancing, and is done in virtually identical format across two languages.

Response: Thank you -- also for the many constructive comments below!

- Reviewer 2: Given the important role that blocks play in the experimental design, it is surprising how little explanation is given as to why these are important and what is innovative about the block design in the current experiment. An attentive reader can piece things together, but the reader really shouldn't have to do this.

Response: We expanded on the rationale for and description of including block of trials in the counterbalancing scheme in Introduction (p. 15) and Methods (p. 19) of Exp 1. Please see also Response to Reviewer 1 above. 

- Reviewer 2: The empirical focus of the satiation experiment is the superiority effect, but very little explanation is given as to why this phenomenon was chosen, beyond the statement that it has "a high degree of variability" (but "variability" in what sense?). Since one of the most interesting claims in the literature is that only certain structures will exhibit satiation, the question of why this structure in particular was chosen deserves more discussion.

Response: We laid out our reasons for choosing multiple questions and the superiority effect in the Introduction (p. 11) of the revised ms. 

- Reviewer 2: The discussion of the materials for the experiment is especially hard to figure out. Partly, this is because the word "sentence" is used sometimes to refer to an actual sentence and sometimes to a set of sentences (what is sometimes called in other papers a "lexicalization set" or "token set"), which means the reader often has to guess what is intended. The numbers of stimuli are also hard to figure out. At one point, it is stated that each block consists of 48 (4 x 12) experimental items and 24 filler items, but then somehow the total number of items in a block is 62. Getting these details right is important, because the reader needs them understand what the experimental-filler ratio is. It is not separately given, but there would appear to very few fillers relative to the experimentals. Once again, though, the reader should not have to be the one figuring this out. The ratio used should be explicitly stated and justified.

Response: The target-filler ratio is 1:2. We reduced the technical terminology and revised the description of the counterbalancing scheme in Introduction (pp. 15-16) and Methods (p. 19) of Exp 1).

- Reviewer 2: As mentioned, blocks are a crucial design feature of this experiment, but it is not made clear what role they play while stimuli are presented to participants. The paper states that participants may rest for as long as they want between blocks, but it isn't clear how they know when each block finishes (given that in other studies, divisions between blocks are invisible to participants). It also isn't stated how long the experiment took, which is potentially important, since the number of stimuli is significantly longer than standard acceptability experiments, which raises possible questions of fatigue, attention to task, etc.

Response: The information is provided in Procedure (p. 20) in the Revision.

- Reviewer 2: At the very end of the paper, the distinction between strong and weak islands is brought up for the first time without any context. The authors seem to suggest that their results have something to say about this distinction, but it isn't clear what it is.

Response: We deleted this paragraph

- Reviewer 2: At several points in the paper, the expression "as for" is used when it appears that "as with" was what was actually intended (e.g., "as for German" when the context makes "as with German" more plausible).

Response: Corrected

REVIEWER 3

- Reviewer 3: The author(s) report the results from three acceptability judgment experiments testing the "exposure" effects, i.e., whether less than fully acceptable sentential structures would get higher improved higher ratings after the participants were repeatedly exposed to them. There has been a debate on whether such an "exposure" effect, if any, is tapping specific grammatical competence (i.e., whether a certain construction can be "satiated"). The author(s) carefully designed questionnaire experiments, where, unlike most of the previous studies, a considerable amount of target and filler items were properly counterbalanced, and found that the improvement of the ratings by "exposure" was found for the items with "middle ratings", irrespective of particular constructions. The author(s) conclude that under a properly designed judgment experiment, the "exposure" effect will show no sensibility to particular grammatical knowledge.

 I think this is a well-controlled study, convincingly arguing that the "satiation" effects, or as the author(s) put it, the exposure effects, are not really sensitive to the type of constructions, but are sensitive to the degree of acceptability (i.e., the middle-rated materials get improved with exposure). I agree with the authors' criticisms on the previous studies on the "satiation" effects, as many of them are actually rather poorly designed, and I am glad to see proper experimental designs testing this issue.

Response: Thank you -- also for the many constructive comments below!

- Reviewer 3: First, the literature review (pp.2-13) seems comprehensive, but it is rather disorganized and very hard to follow. I think this part could be better organized. 

Response: We reorganized and rewrote large parts of the introduction. Specifically, we focus on the core message, namely that satiation is closely related to the intermediate nature of acceptability. We use the literature review to highlight the diversity of approaches, both in terms of substantive topics and methodologies, yielding a somewhat incoherent and sometimes contradictory profile of results in this field of research. Against this background, we provide the reasons motivating our use of superiority violations in the experiment and the role of the fillers in this context.

- Reviewer 3: Second, the full list of experimental materials should be given. In L.643, it is stated that "see Supplement for the complete list of target and filler sentences," but I could not find them there.

Response: The supplement is provided at the OSF repository. Sorry for the oversight. 

- Reviewer 3: Third, I see some sloppiness in writing--typographical errors, seemingly wrong (or obscure) numbers, and some grammatical mistakes. I wish the author(s) checked the manuscript more carefully before submitting it to the journal. 

Response: We thoroughly revised the manuscript. 

- Reviewer 3: L.83-84 "individual counterbalancing of a high number of items" What does this mean? What is "individual counterbalancing"?

Response: We expanded on the rationale for and description of including block of trials in the counterbalancing scheme in Introduction (p. 15) and Methods (p. 19) of Exp 1. Please see also Response to Reviewer 1 above.

- Reviewer 3: LL.204-212. Do we really need the detailed data shown here?

Response: We removed the details. 

- Reviewer 3: LL.276-287 The author(s) cite Chomsky's original definition of the Superiority effect, but do we really need this? This definition is, honestly, very hard to digest. Besides, the present manuscript does not really address the theoretical details of the effect. I think the examples shown in LL.284-286 are good enough.

Response: We removed the definition.

- Reviewer 3: Experiment 1; I'm wondering what the breakdown of the items were.

L.333 "120 target and 252 filler items"

L.422 "six blocks of 372 items" (-> better stated as "six blocks, 372 items in total")

OK, but then:

LL.423-424 "Each block consisted of 62 items (4 x 12 target sentences and 24 filler sentences)": 62 x 6 = 372, but 4 x 12 + 24 = 72, not 62!

120 target items / 6 = 20 target items per block?

252 filler items / 6 = 42 filler items per block? 

Where did these figures "4 x 12 target sentences and 24 filler sentences" come from?

Am I missing something? Some clarification is desired.

Response: We revised the description in “Design and Counterbalancing” in Methods (p. 19) of Exp 1. 

- Reviewer 3: Minor points

L.87. "to test and falsify the effect of grammatical construction". Although scientists try to "test" a hypothesis, they do not try to "falsify" it.

LL.155-168. Reference number [15] is repeatedly referred to here, but shouldn't it be [16]? (I looked into [15] but I couldn't find what the author(s) are pointing at. It took me a while before I finally figure out that this may be a typo.)

L.191 "fillers items" -> "filler items"?

L.202 "many studies do not report means" -> "not many of them report means" or "only a few of them report means" etc.

L.205 "m=5.7333s" -> "m=5.7333"?

L.221 "reports were made movement amongst intermediate judgments" What does this sentence mean? 

LL.263-264 "separate the test whether ... from the test whether ..." -> "the test of whether ..."?

LL.288-290 Shouldn't (3) be (5)?

L.378 "the other half at the end of the main clause" Why did the author(s) made the items this way? Explanation is desired.

LL.379-380 "because the goal of stimulus construction to create sentences..."

-> "was to create"?

L.453 "...well-formedness ratings, that is ratings would increase ..." -> "...well-formedness ratings; that is, ratings would increase ..."

L.490-491 (d) "wen-was" -> "was-wer"?

L.563 "was_wen" -> "wen_wer" 

Response: The above problems were all corrected, revised or do no longer apply in the revision.

REVIEWER 4

- Reviewer 4: The paper presents the results from three graded acceptability judgment surveys on multiple question sentences and superiority, one on German, two on English. The design involves balanced and randomized exposure of the same sentence types over six blocks with pauses in between. Overall, the results show that acceptability ratings slightly increase under repeated exposure, and that speakers prefer subject-initial sentences (Who saw what?) over object- initial ones (What did who see?). Furthermore, subject-object order interacts with animacy in German and D-linking in English making object-subject order more acceptable if an animate or D-linked object precedes and inanimate or non-D-linked subject. The judgments on the four target sentence types in all three experiments fall in the middle range of the 7-point scale (app. 3-5, 3-5.5, 3-6), and the set of 6 different filler types span almost the whole scale (app. 1.5-6.5). As the fillers that are rated highest (type A) and the ones rated lowest (type F) do not increase in acceptability between the first block and the average value for blocks 2-5, it is argued that “satiation” affects a “zone of well-formedness” (i.e., the middle range of acceptability) rather than specific types of constructions. This is seen as going against the assumption in much the literature on processing and acceptability, namely that satiation only affects grammatical sentences. 

The study is very interesting and important as it informs an ongoing debate about language processing, parsing, and grammaticality vs. acceptability. I also agree that it would be nice to have a proper standardized experimental design to test for effects of repeated measures (the literature is mixed and some use a binary scale some a continuous one). 

Response: Thank you -- also for the many constructive comments below!

- Reviewer 4: However, the paper could do with a serious revision and it has a large number of unresolved issues. For example, it is not explained what the zone of well-formedness, and it is not clear how the results go against what is actually argued in the literature. Also, some arguments are based on comparisons between controlled target sentences and mean values for conflated sets of uncontrolled fillers; how this licensed needs to be explained. Some sections are almost impossible to understand, and there are many unexplained terms. 

Response: These criticisms were also provided in the PDF. We address them below.

Reviewer 4: P. 9:

 Line 196 & 199. Please explain that “zone of well-formedness” (on p. 4, line 98-99, it’s called “intermediate zone of judgments”) refers to the middle-range of acceptability on a continuous scale, and it would also be helpful (and appropriate) if it was specified exactly what is meant by “zone” (is it 2-6 on the scale from 1 to 7?). 

 P40. Line 927-928 & 929-930: Please specify what is meant by a “range of judgments”. 

Response: We operationalize “intermediate zone of judgments” as ratings between 3 and 5 (inclusive) and the “zone of well-formedness” as ratings of 3 and above.

- Reviewer 4: P. 10, line 202: “Unfortunately, many studies do not report means.”

Are you sure? The study in [25] clearly does (p. 57: table 2 and fig. 2, p. 63: table 4, p. 64: fig. 4), and the same goes for [17], p. 335: table 3, p. 336: table 4. 

Response: We removed the statement and discuss four studies that provide means. 

- Reviewer 4. P. 12: Line 261: “Keeping a block design allows principled points of measurement...” Why? Testing for the effect of repeated exposure with a high number of tokens per type across a single experiment can just as easily control for individual variation, so why is this proposal better? 

Response: Including block into the counterbalancing scheme converts it to a third experimental factor. This implies that there are no correlations between order and the two other syntactic experimental manipulations. Moreover, all three factors are within-subject/within-item factors ensuring high statistical power. 

- Reviewer 4: P. 13: Line 276-287: I think the phenomenon could be explained clearer with a more recent formulation. As it stands, it is almost completely opaque to readers not familiar with the original framework. Also, nothing in this paper hinges on this particular definition of the Superiority Condition. It could just as well have been relativized Minimality, Shortest Move, the Minimal Link Condition, Economy – or simply Locality. 

Response: The definition was removed.

- Reviewer 4: P. 13: Line 292-295: Please explain why you think that “a variability with multiple factors” makes superiority violations (or any other construction) “particularly suitable”. 

Response: The statement was removed.

- Reviewer 4: P. 15. Line 334-335: “Each target sentence was available in four conditions.” That cannot be true. That would be 120*4 sentences. Do you mean that the same scenario / argument structure was presented in four different conditions/sentence types? 

Response: Indeed, there were 120 items which occurred in 2 x 2 within-item/within-subject “scenarios, that is there were 480 different sentences, but every subject rated only one of the four versions. Therefore, in the LMM, Item is included as a random factor with 120 levels. 

- Reviewer 4: P. 16: Are conditions 3 and 4 generally considered ungrammatical or unacceptable in German? P. 17: “...meaning that superiority is violated.” Which is normally unacceptable in German, or? 

Response: We clarified this aspect in the Introduction of Exp 1 in the Revision. We describe Condition 3 as “not fully well-formed”, the other conditions as “well-formed”.

- Reviewer 4: P. 17 Line 386-387: “We then added a sixth level (F)”. Since you use a 7-point scale (line 330), why do you use only 6 filler types if you want the set of fillers to “better reflect the spectrum of grammaticality”? And how do you know that it actually does reflect the full spectrum? Did you check? Did you do a norming study? 

Response: We modeled the construction of our fillers on the five types of fillers and on the examples described and provided in Featherston and added one clearly ungrammatical type. The choice of the material was one of convenience, not intended to match the seven points of the rating scale. It is an interesting proposal to instruct levels of rating explicitly via gradient levels of (un-)grammatical fillers! Our results replicated those of Featherston. The observed means were compatible with our expectations. We did not check and did not carry out a norming study. 

- Reviewer 4:

P. 19. Line 422:

“The experiment comprised six blocks of 372 items.”->

“The experiment comprised of six blocks of 372 items in total.” 

Line 424-425: Explain what is meant by “counterbalancing scheme”.

Line 425: “... ensured that all items were seen equally often across the six blocks...” Don’t you mean that each type was used equally often and each item was seen only once? 

Line 426-427: “...and particularly important when the number of items per block is higher than 1, ...” I don’t understand. Please explain. Clearly the number of items in each block is higher than 1. Otherwise, there would be 372 blocks... 

Line 428: ”Also target sentences were seen equally often in the four conditions.” Do you mean “target sentence types”? 

Line 430: I am not familiar with the term “Williams design”. Please provide a reference. 

Line 433-434: “This design requires a multiple of 24 subjects and determined the total number of 48 subjects.” How can this “determine” the number of subjects? It seems that it requires at least 24 participants, and that you doubled the number to be sure. 

P. 29: Line 674: Explain “counterbalancing”.

Line 680: Explain how participants were instructed to use the full 7-point scale. 

Response: The above comments were addressed in the Revision. Specifically, we expanded on the rationale for and description of including block of trials in the counterbalancing scheme in Introduction (p. 15) and Methods (p. 19) of Exp 1. Please see also Response to Reviewer 1 above.

- Reviewer 4. P. 18: Line 405-411: There seems to be a sharp drop between Level E and Level F. E is interpretable but less acceptable than D (not unacceptable or ungrammatical), whereas F is uninterpretable and unacceptable (which presumably means the same as completely unacceptable and/or ungrammatical). 

Response: Indeed, we had expected there to be such further drop in acceptability between E and F, but this expectation was not borne out by the ratings of the participants. We discuss the unexpected hehaviour of filler type F in the discussion, and attribute it to the less systematic way in which filler class F was constructed.

- Reviewer 4. P.21, “Statistical analysis”

This section needs to be significantly revised. It is almost impossible to follow. Line 454: What is a “Helmert contrast”? Explain and add reference. 

Response: We describe the contrast and add a reference to a comprehensive treatment and tutorial on the topic of contrast coding (Schad et al., 2020). Helmert contrasts are usually not cited (just like Fisher is no longer cited for ANOVA or Student for t-tests), but unfortunately so far we also could not determine who introduced the term and with reference to exactly which publication of Helmert; there are several candidates. We did learn, however, that Robert Friedrich Helmert died in Potsdam in 1917. We will follow up!

- Reviewer 4. P22. LMM

 Line 459: “We selected a complex mixed model...” What do you mean? The full model? Please specify. 

 Line 461-467: “... we started with an LMM including VCs and PCs ...” This entire paragraph is opaque. What exactly was the model? What is meant by “at the boundary”? What do you mean “forcing CPs to zero”?

 Line 477: “As the six types of filler sentences varied between items, ...” Please explain

 Line 479-480: Please explain and add references for “Forcing CPs to zero”, “AIC”, and “BIC criteria”.

 Line 485-485: Please provide a reference for this. 

Response: We assumed some basic familiarity with LMMs, because they are accepted inferential statistics, replacing the traditional F1-/F2 ANOVA statistics (e.g., Bates et al., 2015a, is cited 50,129 times; the paper introducing them to psycholinguistics, Baayen et al., 2008, 6965 times; our strategy of parsimonious model selection, Bates et al., 2015b, is cited 896 times; all stats retrieved from google scholar on 2021-02-14). However, they are not as familiar as ANOVAs and it is bit difficult to decide about the correct level of detail to be included. For the LMMs in this paper, details about VCs and CPs are not of direct relevance for the interpretation of results. We mentioned them only to document that we included VCs and CPs to guard against false positives. Therefore, rather than expanding on tangential technical statistical detail we refer to the online repo where we provide all the details about model selection in the script and the literature. 

- Reviewer 4.

 Figure 1 (and the same applies to figures 2 and 3): It would be very helpful with standard asterisks indicating significant contrasts (satiation effects, horizontally), (*p<.05, **p<.001) (as well as vertically between types). Extracting this information from the text is in no way easy (and it’s not clear that it’s all there). Also, there should be error bars (e.g., ±1 SE). I would also be helpful if the target sentences were labelled “SO aa”, “SO ai”, “OS ia”, and “OS aa” (or similar) to better reflect the word order contrast and the contrast in animacy. 

 P. 22: Line 501-504: It would be helpful to see the p-values as well. 

 P. 23: Line 510-511: “As is clearly visible in the figure, ratings increased less strongly for “wer- wen...”” It seems to be zero, so “less strongly” is misleading.

 Line 521-522: “The condition with mismatched animacy is almost on par with subject-initial conditions, ...”

This seems to be speculation. In fig. 1b, “was_wer” (OS ia) is closer to the lower-ranking “wen_wer” (OS aa) than to the higher-ranking “wer_was” (SO ai) and “wer_wen” (SO aa). Again, visual indication of significant contrasts in the figures would be helpful. 

Response: We are in general agreement that such figures should show error bars. In this case, our arguments do rest more on the similarity of results across the three experiments than on minute differences within one of the experiments. We had tried alternative versions of the figures with error bars, but they are already very busy and destroy the overall pattern of similarity across the three experiments. The current versions are the best ones we could come up with to communicate the similarity in the patterns of results across the three experiments. Moreover, all the detailed information is available; we report all significant p-values in the text and provide all p-values in the three Tables. Therefore, we opted to keep the format of the figures. 

- Reviewer 4: P. 25: Line 566-567: “Possibly, D represents a “hard-core” ungrammatical filler type from the beginning” Do you mean “F”, not “D”? If D is “hard-core” ungrammatical, then how come E behave like a less hard-core type? 

Response: This statement was removed. 

- Reviewer 4: P. 25:Line 573-574: “... and (iii) exposure effects may be a property of intermediate judgements rather than of certain types of syntactic constructions.”

Yes, but see type F. It does not show increased acceptability over time, and it is only example which is starkly ungrammatical. All the other fillers are increasingly unacceptable, but F is not just less acceptable than E. So, the results here are compatible with the assumption that fully acceptable sentence will not show “satiation”, as they can’t get any better, and fully ungrammatical sentences remain ungrammatical. Sentences with intermediate (degraded) acceptability are susceptible to satiation. This is also how I understand refs. [15] and [25], at least. 

Response: We agree with the statement that it is sentences with intermediate acceptability that are susceptible to satiation, but our general discussion also comments on the different behavior of levels E and F, which start out at roughly the same acceptability level but show different dynamics of their acceptability with some delay. 

- Reviewer 4: P. 28-29

Lines 644-666: The sentences in each level are not minimally different. Therefore, it is not appropriate to average across them and speculate on the interpretation of the contrasts, between filler types as well as (and more importantly) between targets and fillers. The only thing which does allow for interpretation is that some of them show a satiation effect. That says nothing about the status of the well-formedness and satiability of the target sentences.

Response: We address this issue in the final part of the General Discussion

- Reviewer 4: Line 853-854: “Previous studies vary between 5 and 7 blocks, each with 1 instance of the crucial constructions (some studies contain multiple islands (up to 6 in total) ...”

What do you mean “Previous studies vary between 5 and 7 blocks”? What do you mean “some studies contain multiple islands (up to 6 in total)”? 

- Reviewer 4: Line 857: “... the results in the literature on the other hand are zooming in just on the initial part of the experiment (the first two blocks).”

This also applies to the present paper. Almost the entire discussion is based on fig. 1b, fig. 2b, and fig. 3b., which all show the difference between block 1 and the mean value for blocks 2-6 – in effect, the first two blocks. If the point is that there is little or no significant change after block 2, then please make it explicit. Or am I missing something here? 

Response to the two comments above: This part was rewritten. We hope the revision takes care of the concerns.

- Reviewer 4: P. 37-38:

Line 858-867: “First, one would like to know if repeated exposure eliminates the perception of non-well-formedness for certain constructions and for certain grammatical problems, or whether the penalty for violating certain laws of language is merely mitigated. The first interpretation may suggest itself if a certain violation is thought to be a result of the type of processing difficulty proposed in [18] that should no longer exist after a sufficient amount of exposure. In contrast, the second interpretation would be a natural consequence of the view developed in [20] for instance, according to which satiation effects result from improved comprehension strategies (while the perception of grammatical problems would be left intact).” 

 First, repeated exposure does not necessarily “eliminate the perception of non-well- formedness” such that the “processing difficulty” associated with “no longer exist after a sufficient amount of exposure”. Rather, what seems to be argued in [18] as well as in [25] and others is that processing difficulty can be reduced. It doesn’t have to be eliminated. In fact, it is not clear to me why it should be. For example, if the heavy load on working memory can be mitigated with repeated exposure, say with center-embeddings or with certain types of island violations, it doesn’t follow that the participants in the experiments would get massively improved syntactic working memory capacities. It might be improved to a certain point, though (as with other types of memory training). Maybe this is a middle way between the two “interpretations” presented here. The second interpretation, I think, may also be too strong. I agree that there may be alternative or “improved comprehension strategies” involved, but I don’t think that “the perception of grammatical problems” need be “left intact”, because there may not be any “grammatical problems”. Perhaps the reduced / intermediate acceptability rating reflects processing difficulty, as is indeed the case for unambiguously grammatical long-extraction (e.g., crossing two or more clause boundaries) and arguably for certain island violations [25]. 

Response: We reworked this section (see “Dynamics of satiation” in General Discussion) and now remain somewhat agnostic as to the causes of satiation, but slightly favor and give credit to the “mere exposure effect” explanation provided by Zajonc (1968). 

- Reviewer 4: Page 38-39

Line 883-890: “... by looking at where response times destabilise. .... From a visual examination of the log-transformed response times.... Notably, response times stabilised...” You have not included response time measurements in results. (And for experiments 1 and 2, it is not clear how it could have been measured.) 

Response: Correct. We had looked at RTs in an exploratory fashion. However, given that we did not instruct subject to respond fast, their value is quite questionable; for the online experiment they are even less reliable. We removed reference to them, but they are provided in the OSF repository for exploratory analyses. 

- Reviewer 4 : Line 905: “The hope in some of the generative literature [8] that repeated exposure may make questionable sentences (nearly) completely acceptable may be unjustified in any event.”

As in my comment for line 858-867, I think this is too strong. The idea in the literature is not (necessarily) that “questionable sentences” (with reduced acceptability) can be made “(nearly) completely acceptable”. Only that their acceptability can be improved over time. Furthermore, I don’t think that the word “hope” is appropriate here. 

Response: These passages were removed in the Revision 

- Reviewer 4: P. 40. Line 935-936: “So, we observe upward movements of well-formedness for items that may have an unusual object choice, crossing movement, and an ordering constraint. There is no obvious common factor to these.”

 There doesn’t have to be any “obvious common factor to these”. These sentences are all fillers and (as far as has been described in the paper, at least) you did not control for any of the factors you mention here. You conflate and average across different sentences types (and violation types) to get different acceptability ratings along the scale, which makes actual comparison between filler levels and between filler and targets inappropriate.

 Line 937-942: The same applies to English. Indeed, verb-object adjacency can be sometimes be violated, and sometimes it must be violated (‘I saw _ yesterday what might be the ugliest building in the World’), and it’s conceivable that perceived intonation of deviant sentences might facilitate a ‘heavy NP-shift’ reading. 

Response: These points have been taken up in the general discussion of the paper, in particular with respect to the feasibility of using fillers in the way we did. 

- Reviewer 4: Minor points

P. 4: Line 86 and 89 (and in many other places): There are references to numbered sections (e.g., 1.1 and 1.2), but the section headings are not numbered. 

P. 8: Line 150: “In Norwegian...”->“In Danish...” The study in ref. [25] is about Danish, not Norwegian. 

P. 10: 

Line 204-212: What is scale used for the reported means? They are uninterpretable with this information. 

Line 218-219: I don’t understand this sentence. Please reformulate. 

Line 221: “already in the 80s, reports were made movement amongst intermediate judgments [6]” This sentence does not make sense. Please reformulate. 

Line 225: “The mean ratings are given in (3), along with their status with regard to well- formedness.” (a) What is the scale behind the means? (b) The is not indication of their well- formedness (apart from the means). 

Line 272-273: “...condition in (3)...given in (4):”->“...condition in (4)...given in (5):” 

P. 13

Line 288:

”In the sentences in (3), the subject who is superior to (the object what, but ...”-> ”In the sentences in (5), the subject who is superior to (the object what), but ...” (Note also the bracket after “what”.) 

Line 289-290: “... (3a) ... (3b) ...”->“... (5a) ... (5b) ...”

Line 306: I haven’t come across the term “yoking” before. It seems to be an idiosyncratic term used in [31]. Please explain. 

P. 15: 

Line 330: “... ratings from “1” to “7”...”

Did you label the points on the scale? That seems to be hinted at in lines 382-384 below. 

P. 17:

Line 377: “One half of the target sentences had, as in the example above...” Which example? 

Line 377-378: “... an adverb at the beginning of the main clause and the other half at the end of the main clause.” Why this variation? Please give examples of both. 

Line 379-381: There’s a verb missing...

“... because the goal of stimulus construction WAS to create ...” 

Line 381: “and it was certain that the use of adverbs would contribute significantly to this.” How can that be certain? Maybe you should say “assumed” instead. 

Line 382 (and elsewhere): “distractor”/“filler”. Throughout you use the term “filler” far more often than “distractor”. 

Line 382-383: “For the first five gradations from (A) "almost not well-formed" to (E) "completely well-formed", ...”

(i) This seems to be in the wrong order: It should be (A) “complete wellformed” and (E) “almost not wellformed”. (ii) This is not the rating terms used in (7) below, where (A) is defined as “Interpretable and highly acceptable”, and (E) is “Interpretable but less acceptable than (D)”. (iii) There should be an explanation that these labels refer to numerical values on the 7-point scale. 

P. 19:

Line 419: “... no full translation of the whole sentence to English.” “... no full translation of the whole sentence into English.” 

Line 436: “After providing informed consent and collection of demographic information...” Which demographic information? 

Line 438-439: Participants were asked to “to use the full spectrum of the seven-point scale”. How was this done? 

P. 23-24:

Line 532-534: I don’t understand. Please rephrase. 

Line 543: “...there was no interpretable change ...” Do you mean “significant”?

Line 545-546: “Figure 1b exhibits the corresponding pattern of mean changes between block 

1 and block 2.”->“... between block 1 and the mean of blocks 2-6.” 

P. 25: Line 562-563: “For the two object-initial targets we observe higher rating (b=0.40, z=4.24) for “was_wer” than “was_wen”, ...” The “was_wen” condition is subject-initial.

P. 26:

Line 599: Did you make sure that the participants were native speakers of English?

P. 31:

Fig. 2: See my comments to fig 1. above. 

Line 707-710: Are you talking about an overall average (across all types) or about one of the types? 

P. 32:

Line 722: “... (see Figure 2)”->“... (see Figure 2b)” 

Line 723:

What is meant by “the prototype pattern”? 

Line 724-726: These stats are different from the ones in line 713-714. Am I missing something? 

Line 729: “respectively”? 

Line 729-730: What is meant by “statistically parallel”? 

Line 735:

“In summary, as for German target sentences, there is ...”->

“In summary, in English target sentences, just like in the German experiment, there is ...” 

P. 33:

Line 745: “We chose to run the Experiment 2 ...”->“We chose to run Experiment 2 ...” 

Line 783: Explain “use the full spectrum of the seven-point scale”. 

P. 35

See my comments about fig. 1 above. 

P. 36:

Line 830-836: The numbering is wrong. It should be a-d, not g-j. 

Line 831: “A shift in the intermediate zone...” So, do you take “intermediate to mean 1<x<7 or 2<x<6? None of the types are either 1 or 7... 

P. 37:

Line 838: “... violations with like animacy ...”? 

Line 850:

“Our experiments provide strong evidence for an increase, ...”

“Our experiments provide strong evidence for an increase in mean acceptability rating, ...” 

Line 852: “we have 12 per block, i.e. 48 target items per block.” I don’t understand. Please rephrase. 

P. 40:

Line 921: Again, I think the word “hope” is inappropriate. 

Response: All the above stylistic proposals or necessary correction of typos were taken care or do no longer apply to the Revision.

---

## [Decision Letter · Decision Letter 1]

28 Mar 2021

PONE-D-20-32419R1

Middle ratings rise regardless of grammatical construction: Testing syntactic variability in a repeated exposure paradigm

PLOS ONE

Dear Dr. Brown,

Thank you for submitting your manuscript to PLOS ONE. After careful consideration, we feel that it has merit but does not fully meet PLOS ONE’s publication criteria as it currently stands. Therefore, we invite you to submit a revised version of the manuscript that addresses the points raised during the review process.

The paper is almost acceptable, and we would like you to make a few necessary revisions suggested by the reviewer. It should not take too long.

We look forward to receiving your revised manuscript.

Kind regards,

Masatoshi Koizumi, Ph.D.

Academic Editor

PLOS ONE

Journal Requirements:

Reviewers' comments:

Reviewer's Responses to Questions

**Comments to the Author**

1. If the authors have adequately addressed your comments raised in a previous round of review and you feel that this manuscript is now acceptable for publication, you may indicate that here to bypass the “Comments to the Author” section, enter your conflict of interest statement in the “Confidential to Editor” section, and submit your "Accept" recommendation.

Reviewer #4: All comments have been addressed

2. Is the manuscript technically sound, and do the data support the conclusions?

Reviewer #4: Yes

3. Has the statistical analysis been performed appropriately and rigorously? 

Reviewer #4: Yes

4. Have the authors made all data underlying the findings in their manuscript fully available?

Reviewer #4: Yes

5. Is the manuscript presented in an intelligible fashion and written in standard English?

Reviewer #4: Yes

6. Review Comments to the Author

Reviewer #4: The paper benefitted significantly from revision and it is now a very good article.

The authors have addressed all the issues that I raised in my review, and I only have one major comment and a list of minor comments/corrections.

Major:

It would be nice if you could add black dotted horizontal lines to indicate the satiation zone to the plots in Fig1b, Fig2b, and Fig3b. It would also be very helpful if you would add an asterisk for example to the immediate right of the lines for significant changes in Fig1b, Fig2b, and Fig3b. These small additions would be easy to make and would not clutter the plots, and they would make it much easier to see what the plots actually show – both in terms of significant effects (vs. mere trends) as well as differences between English and German in the range of the intermediate zone of satiation.

Minor comments:

L.23-24:

“A prominent approach in linguistic theory claims that” =>

“A prominent approach in linguistic theory argues that”

L.32-33:

“Second, ratings asymptote below maximum acceptability, that is they do not satiate.” =>

“Second, though there is satiation, ratings asymptote below maximum acceptability.”

As is also argued in the paper, the increase in acceptability as a function of exposure IS satiation, and satiation does not necessarily need to lead to full acceptability.

L.67:

“fully inacceptable” =>

“fully unacceptable”

L.98

“illustrated in (1). in which” =>

“illustrated in (1) in which”

L.121, table 1 (Goodall (2011), Adjunct island):

“nos” => “no”

L.156:

“the Complex Noun Phrase Constraint. (1e) violations” =>

“the Complex Noun Phrase Constraint (1e) violations”

L.157:

“not, e.g., that-trace violations 1c)” =>

“not, e.g., that-trace violations (1c))”

L.175:

“Goodall [14] provides another example for the sensitivity” =>

“Goodall [14] provides another example of the sensitivity”

L.182:

“The variability between experiments visible calls…” =>

“The variability between experiments calls…” =>

L.218, (2b):

Use a trace (t) like in (1) instead of strikethrough.

L.223:

[22] => [23]

L.258:

“Examples for the lower three levels of English”

Please explain the scale, what the levels (C,D,E) mean. The scale is not explained in detail until lines 376-382, where it actually also explained that an extra level is introduced (F). So “lower” is not the right term here.

L.274:

“because German material is” =>

“because the German material is”

L.277:

“because of the sentence does not respect” =>

“because the sentence does not respect”

L.342-3:

“(A) "almost not well-formed" to (E) "completely well-formed".”

“(A) "completely well-formed" to (E) " almost not well-formed ".”

L.360:

“[31] design” =>

“design [31]”

L.385:

“The material consisted of 120 targets …” =>

“The material consisted of 120 target quadruples (corresponding to (3) above)” …

L.385:

“… and 252 fillers. that is the ratio of targets” =>

“…and 252 fillers. That is, the ratio of targets”

L.447:

“Williams [31] design” =>

“Williams design [31]”

L.501:

“block 2 did not differ (C1 to C5 [39].” =>

“block 2 did not differ (C1 to C5) [39].”

What is “C1 to C5”? Presumably “C1 to C5” refers to “five of these nine block x type-of-sentences contrasts”, but that is not clear. Since, the details have been moved to the OSF repository. This bit should be left out here. Also, it’s not clear why we need a reference to [39] here.

532-4:

“As is clearly visible in Fig 1, ratings increased less strongly for “wer-wen” and more strongly for “wer-was” constructions than the other three conditions.”

It is NOT “clear”. It looks very much as if “wer_was” and “wen_wer” increase equally much.

L.616:

“generative literature.” =>

“generative literature).”

L.621 & L.627:

“non-discourse-linked object” =>

“non-discourse-linked wh-phrases”

Both subject and object are non-D-linked.

L.757:

“(both z<1.).” =>

“(both |z|<1).”

L.854-5:

“Just as in the preceding two experiments, filler level F behaved in an unpredicted way by not showing any signs of rising judgments.”

Unless I’m missing something, the behavior of F is not unexpected. Since it is tailored to be fully ungrammatical and hence be below the intermediate zone, it is predicted to not show satiation. It’s the behavior of E that is unexpected.

L.948:

“Zajonc [46]” =>

“Zajonc [45]”

L.961

“Zajonc [46]” =>

“Zajonc [45]”

L.962:

“Häussler [47]” =>

“Häussler [46]”

L.970:

“Sprouse et al., [48]” =>

“Sprouse et al., [47]”

L.974:

“variants of targets However, the fillers” =>

“variants of targets. However, the fillers”

7. PLOS authors have the option to publish the peer review history of their article (what does this mean?). If published, this will include your full peer review and any attached files.

Reviewer #4: **Yes: **Ken Ramshøj Christensen

---

## [Author Response · Author response to Decision Letter 1]

16 Apr 2021

Major comment:

• Comment:

It would be nice if you could add black dotted horizontal lines to indicate the satiation zone to the plots in Fig1b, Fig2b, and Fig3b. It would also be very helpful if you would add an asterisk for example to the immediate right of the lines for significant changes in Fig1b, Fig2b, and Fig3b. These small additions would be easy to make and would not clutter the plots, and they would make it much easier to see what the plots actually show – both in terms of significant effects (vs. mere trends) as well as differences between English and German in the range of the intermediate zone of satiation.

Reply: We very much appreciate this proposal, changed the figures accordingly, and agree that it substantially improves what we want to communicate with them. The adoption of this proposal necessitated a change in the specification of the second LMM for each experiment such that the change from block 1 to the average of block 2 to 6 is tested with block nested within levels of sentence type, not as crossed with contrasts defined for types. These statistics are definitely much more comprehensible than the prior ones relating to interactions between contrast and block because with this parameterization asterisks in the figures correspond directly to estimates in the post-hoc LMM. As a side effect we also corrected an error in the labelling of types with a very similar profile of acceptability in the figure of Experiment 3. We note that this was not our initial plan for data analysis, but we consider it acceptable to switch from one post-hoc LMM to a different one. This change required changes in the methods and results. We also include tables with fixed-effects in the supporting material. Again, thank you very much for this comment! 

Minor comments:

• Comment:

L.23-24:

“A prominent approach in linguistic theory claims that” =>

“A prominent approach in linguistic theory argues that”

Reply: changed

• Comment:

L.32-33:

“Second, ratings asymptote below maximum acceptability, that is they do not satiate.” =>

“Second, though there is satiation, ratings asymptote below maximum acceptability.”

As is also argued in the paper, the increase in acceptability as a function of exposure IS satiation, and satiation does not necessarily need to lead to full acceptability.

Reply: changed

• Comment:

L.67:

“fully inacceptable” =>

“fully unacceptable”

Reply: changed

• Comment:

L.98

“illustrated in (1). in which” =>

“illustrated in (1) in which”

Reply: changed

• Comment:

L.121, table 1 (Goodall (2011), Adjunct island):

“nos” => “no”

Reply: changed

• Comment:

L.156:

“the Complex Noun Phrase Constraint. (1e) violations” =>

“the Complex Noun Phrase Constraint (1e) violations”

Reply: changed

• Comment:

L.157:

“not, e.g., that-trace violations 1c)” =>

“not, e.g., that-trace violations (1c))”

Reply: changed

• Comment:

L.175:

“Goodall [14] provides another example for the sensitivity” =>

“Goodall [14] provides another example of the sensitivity”

Reply: changed

• Comment:

L.182:

“The variability between experiments visible calls…” =>

“The variability between experiments calls…” =>

Reply: changed

• Comment:

L.218, (2b):

Use a trace (t) like in (1) instead of strikethrough.

Reply: changed

• Comment:

L.223:

[22] => [23]

Reply: changed

• Comment:

L.258:

“Examples for the lower three levels of English”

Please explain the scale, what the levels (C,D,E) mean. The scale is not explained in detail until lines 376-382, where it actually also explained that an extra level is introduced (F). So “lower” is not the right term here.

Reply: changed (changed “lower” to “intermediate English levels”)

• Comment:

L.274:

“because German material is” =>

“because the German material is”

Reply: changed

• Comment:

L.277:

“because of the sentence does not respect” =>

“because the sentence does not respect”

Reply: changed

• Comment:

L.342-3:

“(A) "almost not well-formed" to (E) "completely well-formed".”

“(A) "completely well-formed" to (E) " almost not well-formed ".”

Reply: changed

• Comment:

L.360:

“[31] design” =>

“design [31]”

Reply: changed

• Comment:

L.385:

“The material consisted of 120 targets …” =>

“The material consisted of 120 target quadruples (corresponding to (3) above)” …

Reply: changed

• Comment:

L.385:

“… and 252 fillers. that is the ratio of targets” =>

“…and 252 fillers. That is, the ratio of targets”

Reply: changed

• Comment:

L.447:

“Williams [31] design” =>

“Williams design [31]”

Reply: changed

• Comment:

L.501:

“block 2 did not differ (C1 to C5 [39].” =>

“block 2 did not differ (C1 to C5) [39].”

What is “C1 to C5”? Presumably “C1 to C5” refers to “five of these nine block x type-of-sentences contrasts”, but that is not clear. Since, the details have been moved to the OSF repository. This bit should be left out here. Also, it’s not clear why we need a reference to [39] here.

Reply: C1 to C5 referred to contrasts defined for sentence types. They are no longer the part of the post-hoc LMM where we test block as nested within type of sentence.

• Comment:

532-4:

“As is clearly visible in Fig 1, ratings increased less strongly for “wer-wen” and more strongly for “wer-was” constructions than the other three conditions.”

It is NOT “clear”. It looks very much as if “wer_was” and “wen_wer” increase equally much.

Reply: the reviewer mentions “wer_was” where we mentioned “wer-wen”. We think the misunderstanding may come from the fact that wer-wen does not increase at all. We were intending to highlight the differences between conditions here (i.e., the contrasts). We no longer have a test of this interaction contrasting “wer_wen” with the average of the other three conditions. The simple tests yielded there is no change for “wer_wen”, but significant changes for the three other types of target conditions.

• Comment:

L.616:

“generative literature.” =>

“generative literature).”

Reply: changed

• Comment:

L.621 & L.627:

“non-discourse-linked object” =>

“non-discourse-linked wh-phrases”

Both subject and object are non-D-linked.

Reply: changed

• Comment:

L.757:

“(both z<1.).” =>

“(both |z|<1).”

Reply: changed

• Comment:

L.854-5:

“Just as in the preceding two experiments, filler level F behaved in an unpredicted way by not showing any signs of rising judgments.”

Unless I’m missing something, the behavior of F is not unexpected. Since it is tailored to be fully ungrammatical and hence be below the intermediate zone, it is predicted to not show satiation. It’s the behavior of E that is unexpected.

Reply: We agree with this comment. In the Discussion we clarify that the rise of E most likely relates to a delayed recognition of less severe ungrammaticality than F (as intended). This should not be confused with a rise of acceptability of these sentences; they are still judged ungrammatical.

• Comment:

L.948:

“Zajonc [46]” =>

“Zajonc [45]”

Reply: changed

• Comment:

L.961

“Zajonc [46]” =>

“Zajonc [45]”

Reply: changed

• Comment:

L.962:

“Häussler [47]” =>

“Häussler [46]”

Reply: changed

• Comment:

L.970:

“Sprouse et al., [48]” =>

“Sprouse et al., [47]”

Reply: changed

• Comment:

L.974:

“variants of targets However, the fillers” =>

“variants of targets. However, the fillers”

Reply: changed

---

## [Editor Report · Decision Letter 2]

23 Apr 2021

Middle ratings rise regardless of grammatical construction: Testing syntactic variability in a repeated exposure paradigm

PONE-D-20-32419R2

Dear Dr. Brown,

We’re pleased to inform you that your manuscript has been judged scientifically suitable for publication and will be formally accepted for publication once it meets all outstanding technical requirements.

Kind regards,

Masatoshi Koizumi, Ph.D.

Academic Editor

PLOS ONE
---

## [Editor Report · Acceptance letter]

29 Apr 2021

PONE-D-20-32419R2 

Middle ratings rise regardless of grammatical construction: Testing syntactic variability in a repeated exposure paradigm 

Dear Dr. Brown:

I'm pleased to inform you that your manuscript has been deemed suitable for publication in PLOS ONE. Congratulations! Your manuscript is now with our production department. 

Kind regards, 

on behalf of

Professor Masatoshi Koizumi 

Academic Editor

PLOS ONE